

# Phase diagram of an extended parafermion chain

Jurriaan Wouters[1*], Fabian Hassler[2†], Hosho Katsura[3,4,5‡] and Dirk Schuricht[1∘]

**1** Institute for Theoretical Physics, Center for Extreme Matter and Emergent Phenomena,
Utrecht University, Princetonplein 5, 3584 CE Utrecht, The Netherlands
**2** JARA-Institute for Quantum Information,
RWTH Aachen University, 52056 Aachen, Germany
**3** Department of Physics, Graduate School of Science, The University of Tokyo,
7-3-1, Hongo, Bunkyo-ku, Tokyo, 113-0033, Japan
**4** Institute for Physics of Intelligence, The University of Tokyo,
7-3-1, Hongo, Bunkyo-ku, Tokyo, 113-0033, Japan
**5** Trans-scale Quantum Science Institute, The University of Tokyo,
7-3-1, Hongo, Bunkyo-ku, Tokyo, 113-0033, Japan

⋆ j.j.wouters@uu.nl     † hassler@physik.rwth-aachen.de
‡ katsura@phys.s.u-tokyo.ac.jp     ∘ d.schuricht@uu.nl

## Abstract

We study the phase diagram of an extended parafermion chain, which, in addition to terms coupling parafermions on neighbouring sites, also possesses terms involving four sites. Via a Fradkin–Kadanoff transformation the parafermion chain is shown to be equivalent to the non-chiral $\mathbb{Z}_3$ axial next-nearest neighbour Potts model. We discuss a possible experimental realisation using hetero-nanostructures. The phase diagram contains several gapped phases, including a topological phase where the system possesses three (nearly) degenerate ground states, and a gapless Luttinger-liquid phase.

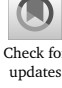

# 1 Introduction

The properties, experimental realisations and potential applications of Majorana fermions in condensed-matter systems have been studied to a great extent in the past two decades. In a seminal work Kitaev [1] introduced, amongst other things, a one-dimensional toy model of spinless fermions and showed that the phase diagram contained a topological phase where Majorana zero modes are localised at the edges. The Majorana chain is equivalent to the well-known quantum Ising chain (see, eg, Fendley [2]). The topological and trivial phases of the Majorana chain correspond to the ferromagnetic and paramagnetic phases of the Ising model, separated by a transition described by a conformal field theory (CFT) [3,4] with central charge $c = 1/2$. Several extensions of this toy model have been studied, like the inclusion of disorder [5–7], interactions [8–14], or both [15–18]. Without disorder, the interacting Majorana chain is equivalent to the axial next-nearest neighbour Ising (ANNNI) model [19, 20]. Besides the topological and trivial phases, already present in the absence of interactions, this model also possesses an incommensurate charge density wave phase as well as a Mott insulating phase [10,21–24]. The Majorana zero modes were thought to find use in topological quantum computation, however, it turns out they are not sufficient to implement universal quantum gates [25, 26].

The Majorana/quantum Ising chain possesses a $\mathbb{Z}_2$-symmetry. An obvious path for generalisation is given by considering $\mathbb{Z}_3$-symmetric[1] systems, which in turn leads to parafermions [27].

---

[1]The generalisation to arbitrary $\mathbb{Z}_n$-symmetry is straightforward, however, in this article we will restrict ourselves to $n = 3$.

In the corresponding parafermion chain the $\mathbb{Z}_3$-symmetry turns out to be less restrictive than the $\mathbb{Z}_2$-symmetry of its Majorana cousin. For example, the breaking of time-reversal and spatial parity symmetry via chiral interactions is allowed. The parafermion chain is equivalent [2] to the $\mathbb{Z}_3$-clock model, which, in the non-chiral case, simplifies to the three-state quantum Potts chain [28]. The latter possesses an ordered phase with three-fold degenerate ground state, which is separated from a paramagnetic phase by a quantum phase transition described by a CFT with central charge $c = 4/5$. In addition, the chiral model possesses an incommensurate phase [29–31]. Interestingly, the transition between the ordered and paramagnetic phases in the non-chiral model is no longer described by a CFT [31]. In the parafermion description the ordered phase is topological, possessing zero-energy modes linked to the degeneracy of the ground state [2, 32–34]. As for the Majorana fermions, this degeneracy alone is insufficient for universal quantum computation. However, through the Read–Rezayi state [35, 36] or with the help of the Aharanov–Casher effect [37] universal gates can be realised from parafermion modes. First steps towards the experimental realisation of parafermion excitations have been taken recently [38–40].

In addition to the chiral interactions, the $\mathbb{Z}_3$-symmetry allows several extensions of the parafermion chain, which correspond to the terms coupling parafermions beyond neighbouring sites [41, 42, 44, 45]. The equivalent clock models can be viewed as $\mathbb{Z}_3$-generalisations of the ANNNI model. It is interesting to note that for specific parameters these clock models become frustration free [42, 46], implying that the degenerate ground states can be constructed explicitly. This behaviour generalises the well-known frustration-free Peschel–Emery line [19] of the ANNNI model.

In this work we focus on a specific extension of the parafermion chain, which, in addition to terms coupling parafermions on neighbouring sites, also possesses terms involving four sites next to each other. In terms of clock variables our model becomes the non-chiral $\mathbb{Z}_3$ axial next-nearest neighbour Potts (ANNNP) model [47]. Our specific choice is motivated by a possible experimental realisation of this extended parafermion chain using heterostructures containing ferromagnets, superconductors and fractional quantum Hall states. We provide a detailed characterisation of the phase diagram of our model (shown in Figure 2), which, for moderate strengths of the extension, contains four gapped phases: the topological and trivial phases already present in the pure parafermion chain, and two phases showing antiferromagnetic and ferromagnetic Ising-type order. In addition, we identify a critical Luttinger-liquid phase with central charge $c = 1$. The latter as well as the two Ising-type phases can be linked to the physics of the spin-1/2 XXZ Heisenberg chain. Furthermore, we provide evidence that the topological phase is pinched between the Luttinger-liquid phase and the ferromagnetic Ising phase.

This article is organised as follows: In the next section we define the extended parafermion chain. In Section 3 we discuss a proposal to experimentally realise it in heteronanostructures, thus motivating our specific choice of the considered extension. We then link the extended parafermion chain to the non-chiral ANNNP model, which provides the starting point for our further analysis. In Section 5 we give a qualitative discussion of the phase diagram, whose details are elaborated on in Sections 6 and 7. We then give a brief outlook on the phase diagram at stronger extension parameters, followed by a concluding discussion of our results in Section 9. The appendix contains further details of our analysis, including a discussion of duality transformations, additional supporting numerical results, and details of the mapping to the effective XXZ chain.

## 2 Extended parafermion chain

In this article we are investigating the phase diagram of a one-dimensional parafermionic system which can be viewed as an extension of the parafermion chain [2,32] by terms coupling parafermions on four neighbouring sites. Specifically, we consider an open chain of length $2L$. At each lattice site we define parafermion operators $\chi_l$, $l = 1, \ldots, 2L$, satisfying (we recall that we consider $\mathbb{Z}_3$-symmetric systems only)

$$\chi_l^3 = 1, \quad \chi_l^\dagger = \chi_l^2, \quad \chi_l \chi_m = \omega^{\text{sgn}(m-l)} \chi_m \chi_l \quad \text{for } m \neq l, \qquad \omega = e^{2\pi i/3}, \tag{1}$$

which can be regarded as a direct generalisation of Majorana fermions. Using this the Hamiltonian of the extended parafermion chain can be written as

$$H = -J \sum_{j=1}^{L-1} \chi_{2j} \chi_{2j+1}^\dagger - f \sum_{j=1}^{L} \chi_{2j-1}^\dagger \chi_{2j} + U \sum_{j=1}^{L-1} \chi_{2j-1}^\dagger \chi_{2j} \chi_{2j+1}^\dagger \chi_{2j+2} + \text{h.c.}. \tag{2}$$

The parameters $J$, $f$ and $U$ are assumed to be real[2], making the model non-chiral. Unless it is stated otherwise, we set $J = 1$. In the absence of the last term, ie, $U = 0$, this model is known as the parafermion chain [2, 32]. The term $\propto U$ corresponds to an extension involving four neighbouring sites. One thus might be tempted to call the model (2) "interacting parafermion chain", however, due to the non-trivial relations (1) the model is not quadratically solvable even for $U = 0$. We note that a similar extension to the parafermion chain has been studied by Milsted et al. [43] and Zhang et al. [44]. The former focused on the $\mathbb{Z}_6$-variant of Equation (2), while the latter discussed the $\mathbb{Z}_3$-model in a different parameter regime[3]. For $f = U = 0$ we recognise that $\chi_1$ and $\chi_{2L}$ decouple from the system and form a non-local zero-energy edge mode that generates a three-fold degeneracy throughout the whole spectrum. This degeneracy is protected by the non-local $\mathbb{Z}_3$-symmetry $\omega^P = \prod_j (\chi_{2j-1}^\dagger \chi_{2j})$. Contrary to the Majorana chain, these exact modes disappear when going away from the classical point. While the ground state might retain its degeneracy, the degenerate excited states hybridise and thus split in energy [2,32–34], ie, the zero modes cease to commute with the full Hamiltonian. The region around the classical point where the ground state remains (approximately) degenerate is called the topological phase.

Before analysing the phase diagram of the extended parafermion chain (2), in the next section we present a proposal to experimentally realise the model using heterostructures containing ferromagnets, superconductors and fractional quantum Hall states.

## 3 Proposal for experimental realisation

Recently, there have been several proposals put forward that allow to realise parafermionic bound states by cleverly constraining the fractionalised edge states of two-dimensional interacting systems [25, 36, 48]. To fix the ideas, we discuss the set-up described by Ref. [48] in more detail. As our starting point we consider helical edge states of a fractional quantum spin Hall state at filling factor $\nu = 1/m$, experimentally observed in [38,39]. Such an edge configuration can also be realised at the interface of two fractional quantum Hall states with $g$-factors of opposite signs [25]. Independent of the realisation, the low-energy degrees of freedom are counter-propagating modes of fractionalised electrons with charge $e^* = e/m$ and spin $1/m$ (in units of the electron spin), see Figure 1.

---

[2]Complex parameters would lead to chiral interactions, which in turn break spatial parity and time-reversal symmetry.

[3]The Hamiltonian in Reference [44] is related to Equation (2) via a duality transformation as discussed in Appendix A.



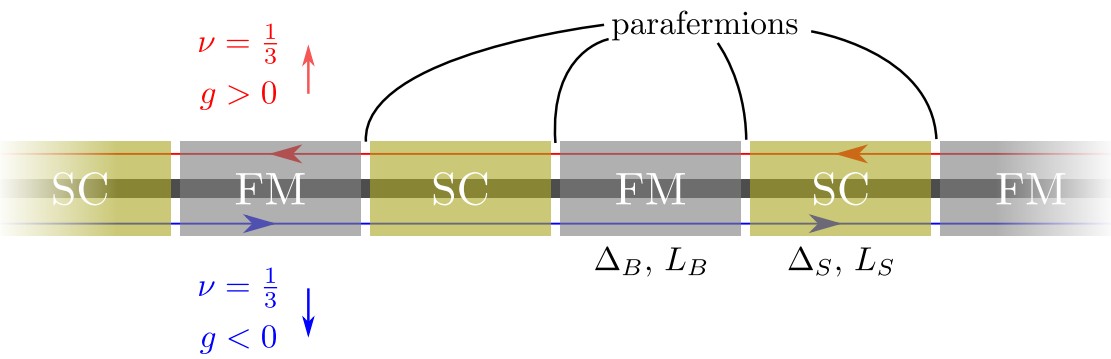

Figure 1: Schematic display of a fractional quantum Hall system with appearing effective parafermion degrees of freedom. The alternating placement of superconductors (SC) and ferromagnets (FM) traps the edge modes. These trapped modes obey the $\mathbb{Z}_6$-parafermion algebra.

There are two (dual) ways of opening a gap in these edge states. Coupling them to a ($s$-wave) superconductor (SC) allows a transfer of charge $2e$ to and from the superconducting condensate. The electric charge $eQ_j$ on the $j$-th superconducting island can thus assume the values

$$eQ_j = 0, \frac{e}{m}, \frac{2e}{m}, \dots, \frac{(2m-1)e}{m} \quad (\text{mod } 2e). \tag{3}$$

The (clock) operator describing the charge is thus given by $e^{i\pi Q_j}$ and commutes with the Hamiltonian [48]. First indications for induced superconductivity in fractional quantum Hall edge states have been reported in Reference [40].

The second way to open a gap is via backscattering. This involves a change of the spin which can be achieved by coupling the edge state to a ferromagnetic (FM) insulator. The spin $S_j$ in the $j$-th ferromagnetic region may assume the values

$$S_j = 0, \frac{1}{m}, \frac{2}{m}, \dots, \frac{(2m-1)}{m} \quad (\text{mod } 2), \tag{4}$$

due to the fact that the ferromagnet serves as a reservoir of spins in units of 2. Note that the backscattering leads to the formation of an insulating phase and correspondingly the charge vanishes in the FM segments. The corresponding clock operators satisfy

$$e^{i\pi S_j} e^{i\pi Q_k} = e^{i\frac{\pi}{m}(\delta_{j,k+1}-\delta_{j,k})} e^{i\pi Q_k} e^{i\pi S_j}, \tag{5}$$

displaying the fractional statistics. Using the algebra in Equation (5) the SC and FM operators can be represented by the parafermion modes on the interfaces,

$$\chi_{2j}\chi_{2j-1}^{\dagger} = e^{i\pi Q_j}, \quad \chi_{2j+1}\chi_{2j}^{\dagger} = e^{i\pi S_j}. \tag{6}$$

With this procedure only $\mathbb{Z}_{2m}$-parafermions can be realised natively while we concentrate on the case $\mathbb{Z}_3$ in this work. Note however that starting from $\mathbb{Z}_6$ ($m = 3$), $\mathbb{Z}_3$-parafermions naturally emerge by allowing for fluctuations of the gauge field with restricted dynamics [51]. An alternative experimental avenue to the $\mathbb{Z}_3$-parafermions is the spin-unpolarised $\nu = 2/3$-state [36].

The entrapment of the parafermions is not perfect and exchange processes through the FMs and SCs couple the parafermions. Tunnelling of a fractional charge $e^*$ through the FM segments is described by the operator $e^{i\pi S_j}$ and yields the term

$$H_J = -J \sum_j \left( e^{i\theta} e^{i\pi S_j} + \text{h.c.} \right), \tag{7}$$

with some coupling $Je^{i\theta}$. As the coupling is due to tunneling of quasiparticles, it is given by $J \propto \exp[-c_B \Delta_B L_B/(\hbar v)]$ and depends exponentially on the length $L_B$ and the gap $\Delta_B$ of the magnetic island, with $v$ the velocity of the edge modes and $c_B \sim 1$ is some constant. A priori, the coupling is complex, but using an instanton approach it can be shown [49] that $\theta = 0$. Moreover, we set $J = 1$, fixing the overall energy scale.

Charging effects on the small mesoscopic islands perturbatively can only involve the operator $e^{i\pi Q_j}$. The charging effects are due to the Aharanov–Casher phase of a superconducting vortex encircling the island [10]. The charging energy assumes the form

$$H_f = -\sum_j \left( f\, e^{i\pi Q_j} + \text{h.c.} \right), \tag{8}$$

where $f \propto \exp[-c_S \Delta_S L_S/(\hbar v)]$, with the length $L_S$ and the gap $\Delta_S$ of the superconducting island, can be made real by an appropriate gate voltage [49, 50]. This term is due to the self-capacitance of the island. The terms (7) and (8) realise the (dual of) $\mathbb{Z}_3$-Potts model studied in Reference [2]. This is the parafermionic analog of the Kitaev chain [1].

Following Reference [10], we argue that the charging effects due to cross-capacitances between adjacent islands are important. They are described by the term

$$H_U = U \sum_j \left( e^{i\pi(Q_j + Q_{j+1})} + \text{h.c.} \right), \tag{9}$$

with $U \in \mathbb{R}$ due to the Aharonov–Casher effect encircling two adjacent islands. We note that the realisation proposed here will generically lead to the regime $|U| \propto \exp[-2c_S \Delta_S L_S/(\hbar v)] \lesssim |f|$. With the relation (6), the effective Hamiltonian $H_J + H_f + H_U$ maps to (2) whose phase diagram we will investigate in the following.

## 4  ANNNP model

The analysis of the phase diagram of the extended parafermion chain will be fostered by mapping it to the equivalent non-chiral $\mathbb{Z}_3$-ANNNP model [47]. The latter generalises the quantum Potts chain by including an additional coupling term, which is reminiscent of the addition of a transverse interaction term when generalising the quantum Ising chain to the ANNNI model [19, 20].

We begin with the Fradkin–Kadanoff transformation [27]

$$\chi_{2j-1} = \left( \prod_{k=1}^{j-1} \tau_k \right) \sigma_j, \quad \chi_{2j} = \left( \prod_{k=1}^{j-1} \tau_k \right) \sigma_j \tau_j = \chi_{2j-1} \tau_j, \tag{10}$$

which relates the $2L$ parafermion operators $\chi_l$ to clock operators $\sigma_j$ and $\tau_j$, $j = 1, \ldots, L$. These clock operators commute off-site,

$$[\tau_i, \tau_j] = [\sigma_i, \sigma_j] = [\tau_i, \sigma_j] = 0, \quad i \neq j, \tag{11}$$

while on the same lattice site they satisfy

$$\sigma_j^3 = \tau_j^3 = 1, \quad \sigma_j^\dagger = \sigma_j^2, \quad \tau_j^\dagger = \tau_j^2, \quad \sigma_j \tau_j = \omega \tau_j \sigma_j, \quad \omega = e^{2\pi i/3}. \tag{12}$$

An explicit matrix representation for the clock operators on an individual lattice site is given by

$$\tau = \begin{pmatrix} 1 & & \\ & \omega & \\ & & \omega^2 \end{pmatrix}, \quad \sigma = \begin{pmatrix} & & 1 \\ 1 & & \\ & 1 & \end{pmatrix}. \tag{13}$$

In terms of the clock operators the extended parafermion chain (2) becomes[4]

$$H = -J \sum_{j=1}^{L-1} \sigma_j \sigma_{j+1}^\dagger - f \sum_{j=1}^{L} \tau_j + U \sum_{j=1}^{L-1} \tau_j \tau_{j+1} + \text{H.c.},\qquad(14)$$

with $J = 1$. We note that the ANNNP model resides on a chain of length $L$, ie, there has been an effective halving of the system size. The non-local $\mathbb{Z}_3$-symmetry[5] of the Hamiltonian is generated by $\omega^P = \prod_j \tau_j$. On the clock variables the spatial parity transformation acts as [30] $\Pi\sigma_j\Pi = \sigma_{L-j+1}, \Pi\tau_j\Pi = \tau_{L-j+1}$, while time reversal is implemented via $T\sigma_j T = \sigma_j$, $T\tau_j T = \tau_j^\dagger$ together with complex conjugation of scalars. This shows that indeed for real parameters $J$, $f$ and $U$ the system is time-reversal and parity invariant. At $U = 0$ the model reduces to the quantum Potts chain [28], which possesses a critical point at $f = 1$ described by a CFT with central charge $c = 4/5$ [3,4]. At $f = 0$ we obtain the classical (ferromagnetic) Potts model, which has a three-fold degenerate ground state.

## 5  Phase diagram

The phase diagram of the extended parafermion chain/$\mathbb{Z}_3$-ANNNP model for weak to moderate values of $U$ is shown in Figure 2. The phases and transitions were studied using a combination of numerical simulations, conformal field theory [3,4] and perturbative arguments. For the numerics we used the TeNPy implementation [52] of the density matrix renormalisation group (DMRG) algorithm [53,54], with a typical bond dimension of 500–1000, unless otherwise stated. First the rough topography of the phase diagram was obtained from an inexpensive DMRG calculation, see Figure 12 of the supporting numerical results in Appendix B. Then the detailed properties of the phases and transitions were investigated, as is discussed in Sections 6 and 7.

We see that the model displays a variety of phases. The top half of the phase diagram ($f \geq 0$) resembles the picture for the ANNNI model [10,12], with two gapped phases separated by a critical line. The ground state of the paramagnetic phase is singly degenerate, while the $\mathbb{Z}_3$-ordered phase has a three-fold degenerate ground state. The latter is due to approximate zero-energy parafermion modes, which explains the term "topological phase". The top half of the phase diagram is discussed in detail in Section 6.

In contrast to the ANNNI model, the $\mathbb{Z}_3$-ANNNP model is not invariant under $f \to -f$ (which is a consequence of the $\mathbb{Z}_2$-symmetry of the ANNNI model). The lack of this invariance is manifest in the phase diagram, which shows four phases for $f < 0$: the topological phase, a gapped antiferromagnetic phase, a critical XXZ phase, and a ferromagnetic phase. The latter three can be related to the physics of the XXZ chain in the limit $f \to -\infty$, which predicts the transitions to be at $U = \pm 1/3$. The detailed description of these phases is presented in Section 7.

## 6  Upper half of the phase diagram ($f \geq 0$)

Given that the $\mathbb{Z}_3$-ANNNP model is not integrable, the applicability of analytical methods is limited. Still, the quantum Potts model ($U = 0$) is well understood due to its relation to the

---

[4]We note that $\mathbb{Z}_3$-symmetry also allows terms like $\propto \tau_j \tau_{j+1}^\dagger$, which we do not consider here. For general $\mathbb{Z}_n$-symmetric models the complexity increases accordingly.

[5]We note in passing that the model (14) possesses an additional $\mathbb{Z}_2$-symmetry $\sigma_j \to \sigma_j^\dagger$, $\tau_j \to \tau_j^\dagger$ which enlarges the $\mathbb{Z}_3$-symmetry to a full $S_3$-symmetry [34].

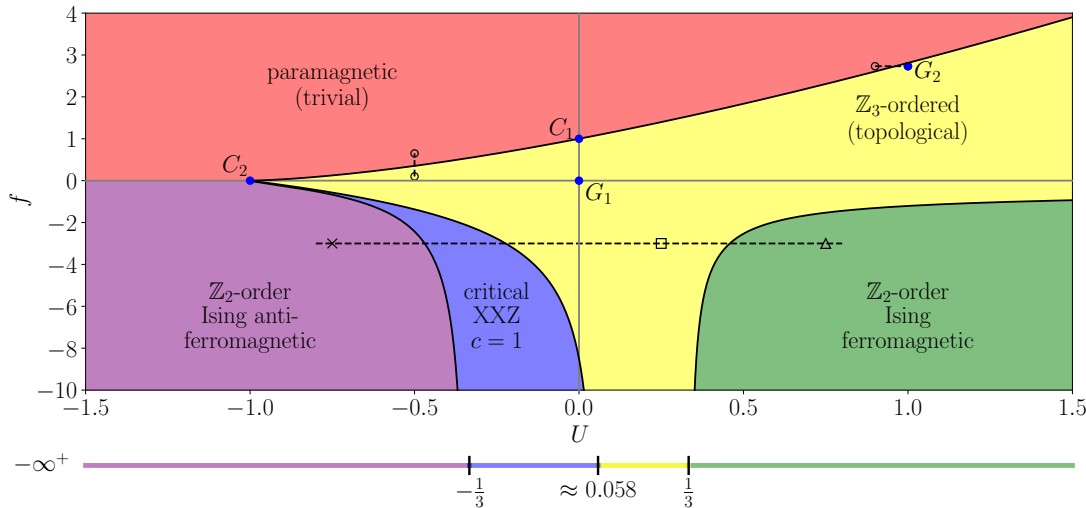

Figure 2: Phase diagram of the extended parafermion chain/$\mathbb{Z}_3$-ANNNP model. We distinguish the following four gapped phases: a paramagnetic phase (red), a topological phase (yellow), an Ising antiferromagnetic (purple) and an Ising ferromagnetic (green) phase. Furthermore, we identify a critical XXZ like phase (violet) with central charge $c = 1$. We also indicate the transition points $C_1$ and $C_2$ corresponding to specific conformal field theories, and the points $G_1$ and $G_2$ at which the model becomes frustration-free and thus allows an exact description of the ground state. At the bottom we show the phase diagram in the limit $f \to -\infty^+$ obtained analytically in Section 7.1. The dashed lines indicate cuts along which detailed results are shown in Figures 3, 4, and 8(b) (with the corresponding symbols for marked points).

two-dimensional classical Potts model. Two topologically distinct phases are separated by a quantum phase transition at $f = 1$ ($C_1$ in Figure 2) described by a CFT with central charge $c = 4/5$. The two distinct phases can be characterised by analysing the limiting cases $f \to \infty$ and $f = 0$, respectively.

In the limit $f \to \infty$ the ground state is unique and given by a product state

$$|\Psi_0\rangle = |0\rangle_\tau^{\otimes L} \,, \tag{15}$$

where $|i\rangle_\tau$, $i = 0, 1, 2$, span the space of eigenstates of $\tau$,

$$\tau |i\rangle_\tau = \omega^i |i\rangle_\tau \,. \tag{16}$$

In the parafermionic language this is identified as the trivial phase due to the absence of boundary modes. The whole phase denoted as paramagnetic in Figure 2 is adiabatically connected to this limit, in particular, it possesses a unique ground state with an energy gap above it. Explicit numerical evidence for the gap at a representative point ($U = -1, f = 1$) is shown in Figure 13(a) of Appendix B.2.

The nature of the $\mathbb{Z}_3$-ordered phase is obvious from studying the $f = 0$ point ($G_1$). Here the three-fold degenerate ground state is given by

$$|\Phi_0^i\rangle = |i\rangle_\sigma^{\otimes L} \quad \text{for } i = 0, 1, 2, \tag{17}$$

where $|i\rangle_\sigma$ span the space of eigenstates of $\sigma$,

$$\sigma |i\rangle_\sigma = \omega^i |i\rangle_\sigma \,. \tag{18}$$

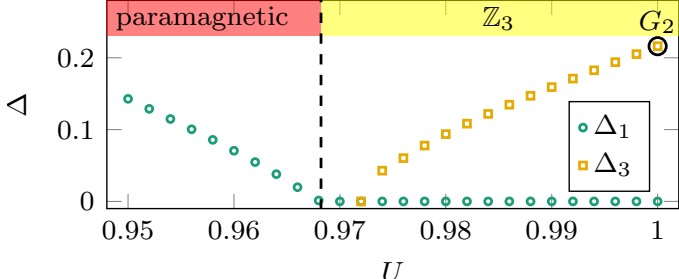

Figure 3: Energy gaps $\Delta_n$ between the ground state and the $n$th eigenstate obtained from finite-size scaling as a function of $U$ for fixed $f = 1+\sqrt{3}$ (see dashed line close to $G_2$ in Figure 2). In the $\mathbb{Z}_3$-ordered phase we observe the three-fold degeneracy of the ground state ($\Delta_1 = \Delta_2 = 0$) with a finite gap above it ($\Delta_3 > 0$). In contrast, in the paramagnetic phase the ground state is unique ($\Delta_1 > 0$). The transition (determined with the methods discussed in Section 6.1) is located at $U_c \approx 0.97$. The gap $\Delta_3$ is very small close to the transition.

The parafermion dual of this system is topological, with edge states $\chi_1$ and $\chi_{2L}$.

Recent progress on frustration-free models allows us to analytically discuss one additional point in the topological phase. In Ref. [42] it was shown that at $U = 1, f = 1+\sqrt{3}$ the model is frustration free (point $G_2$), enabling the construction of the exact ground states. Furthermore, this point is adiabatically (ie, without closing the energy gap) connected to the classical Potts model ($G_1$) [46]. In fact, the points $G_1$ and $G_2$ lie on a frustration free line of a more general Hamiltonian, obtained from (14) by adding a term $\propto (\tau_j \tau_{j+1}^\dagger + \text{H.c.})$ with a suitable prefactor. The situation is reminiscent to the Peschel–Emery line in the ANNNI model [12, 19]. The numerically calculated energy gaps shown in Figure 3 confirm that at $G_2$ the model indeed possesses a three-fold degenerate ground state. The model is gapped down to the transition to the paramagnetic phase at $U_c \approx 0.97$. Further numerical results presented in Appendix B.2 [see Figure 13(b) for the point $U = f = 1$] show that the model is gapped with a three-fold degenerate ground state throughout the topological phase.

Finally, a simplification occurs along the line $f = 0$. Performing two duality transformations (see Appendix A for the details) we can bring the Hamiltonian in the following form

$$H = -\sum_{a=\text{o,e}} \left[ \sum_{j=1}^{L/2-2} \sigma_j^a (\sigma_{j+1}^a)^\dagger - U \sum_{j=1}^{L/2-1} \tau_j^a \right] + \text{H.c.}, \tag{19}$$

where we omitted the boundary terms[6]. The result (19) represents two decoupled (o/e) quantum Potts chains. Consequently, at $U = -1$ the model possesses a second-order phase transition corresponding to a CFT with $c = 4/5 + 4/5 = 8/5$ (see also Reference [55]) depicted by $C_2$ in Figure 2, separating a trivial from a topological phase.

## 6.1 Potts transition in the vicinity of $C_1$

In this subsection we perform a more detailed analysis of the Potts transition. We begin with a scaling analysis of finite-size data, followed by a inspection of the vicinity of the point $C_1$.

---

[6]Here we are only interested in bulk criticality, for which boundary terms can be disregarded. The boundary terms are given in Appendix A.

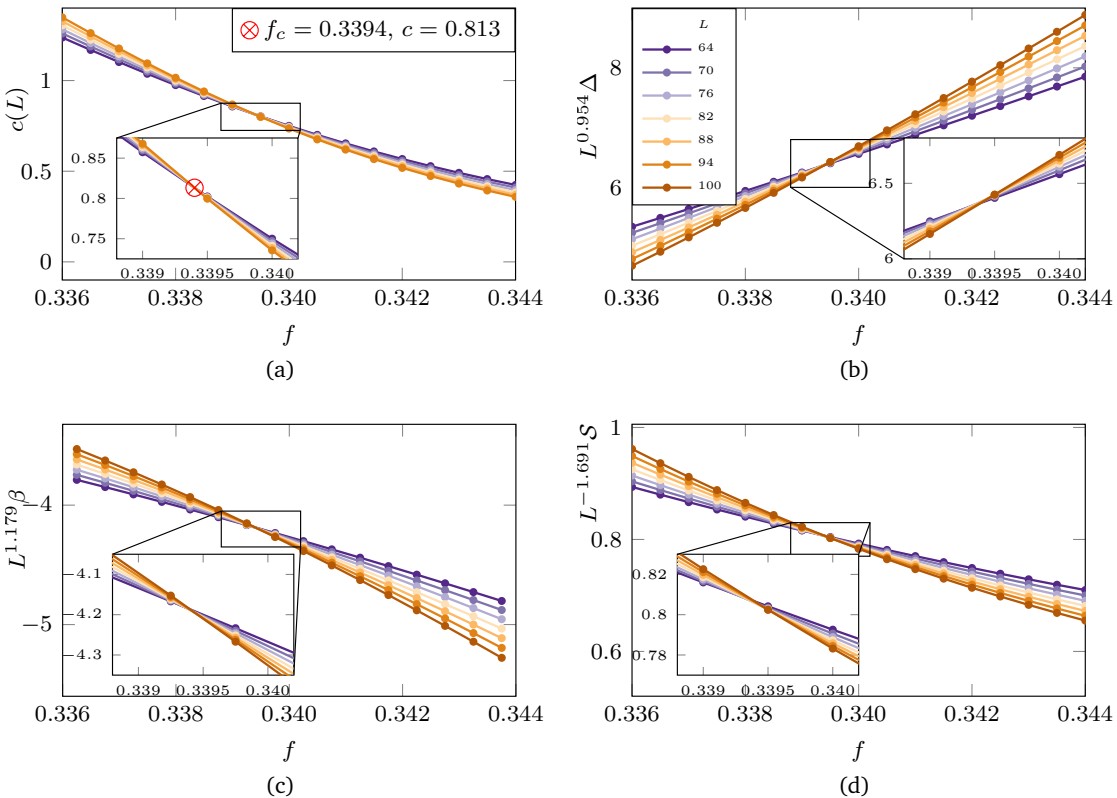

Figure 4: Finite-size results for $U = -0.5$, locating the transition at $f_c = 0.3394$ from the central charge (a) with $c \approx 0.813$ ($L_{\max} = 100$). From (b) we confirm the dynamical exponent is close to 1. From the Callan–Symanzik $\beta$ function in (c) we derive $1/\nu \approx 1.179$ and the structure factor $\mathcal{S}$ gives $2 - \eta = 1.691$. The exponents in (b), (c) and (d) are obtained by requiring that the finite-size data are independent of $L$ at the transition $f_c$.

### 6.1.1 Scaling analysis

In Figure 4 we show several observables along a cut at $U = -0.5$ (indicated by a dashed line in Figure 2). The numerical data were obtained for system sizes $L = 64, 70, \ldots, 100$.

First, we consider the entanglement entropy $S$. In a conformally invariant system this is predicted by the Calabrese–Cardy formula [56,57]

$$S(L, l) = S_0 + \frac{c}{6} \log\left[\frac{L}{\pi} \sin\left(\frac{\pi l}{L}\right)\right],\tag{20}$$

with $c$ being the central charge, $l$ the bipartition length, and $S_0$ being a model-dependent constant. Setting $l = L/2$ we obtain the central cut entanglement entropy, for which we realise that

$$c = 6 \frac{S(L, L/2) - S(L_{\max}, L_{\max}/2)}{\log(L/L_{\max})}.\tag{21}$$

For a critical system the right-hand side of (21) is length ($L$) independent. Thus we can locate the transition as the point where the finite-size data collapse, obtaining the central charge in the process. From the entanglement-entropy results in Figure 4(a) we can infer that $f_c = 0.3394$ with $c \approx 0.813$, which is in good agreement[7] with the predicted value of $c = 4/5$.

---

[7]The numerical results typically improve with an increase in system size and bond dimension.

Second, we consider the energy gap whose scaling behaviour is given by [31,58]

$$\Delta(L) = L^{-z}\tilde{\Delta}(L^{1/\nu}|f - f_c|), \tag{22}$$

with $z$ being the dynamical exponent. The critical exponent $\nu$ governs the divergence of the correlation length $\xi \propto |f - f_c|^{-\nu}$. At the $f = f_c$ we find $z$ by requiring $L^z\Delta(L)$ to be independent of $L$. From this ansatz we obtain $z \approx 0.954$ [see Figure 4(b)], in good agreement with the value $z = 1$ expected for a CFT.

Third, we consider the Callan–Symanzik function $\beta$ [59]

$$\beta = \frac{\Delta}{\Delta - 2\frac{\partial\Delta}{\partial\ln f}} \propto |f - f_c|, \tag{23}$$

which allows us to determine the critical exponent $\nu$. The finite-size ansatz implies that $\beta(L)$ scales as $L^{-1/\nu}$. The CFT prediction for the Potts transition is determined from the scaling dimension of the perturbing field, in this case the energy operator $E$, to be

$$\nu = \frac{1}{2 - \Delta_E} = \frac{5}{6}, \tag{24}$$

with $\Delta_E = 4/5$ for the critical Potts model [4]. From Figure 4(c) we get the numerical value $1/\nu = 1.179$, again close to the prediction.

Finally, the last critical exponent we can easily study is the scaling of the two-point correlation function $\Gamma(r) = \langle\sigma_{i+r}^\dagger\sigma_i\rangle \propto r^{-\eta}$ with $\langle.\rangle$ denoting the ground-state expectation value. From the finite-size scaling ansatz we see that the structure factor behaves as

$$\mathcal{S}(L) = \sum_{i,j}\langle\sigma_i\sigma_j^\dagger\rangle \propto L^{2-\eta}. \tag{25}$$

From the CFT description we recognise that $\eta$ relates to the scaling dimension of the $\sigma$-field [4] $\eta = 4\Delta_\sigma = 4/15$. Consequently, the theoretical prediction is $2 - \eta = 26/15 \approx 1.7333$, with the numerical data in Figure 4(d) yielding the estimate $2 - \eta = 1.691$.

We obtained similar results for several points along the transition line depicted in Figure 2, indicating that the transition along the whole line is described[8] by the Potts CFT with $c = 4/5$.

### 6.1.2 Perturbation around $C_1$

In general it is possible to link the lattice operators in the quantum Potts chain to scaling fields in the Potts CFT [60]. Unfortunately, for the $\tau_j\tau_{j+1}$-perturbation coupled to $U$, which is of interest here, the corresponding field expansion was not derived in Reference [60]. However, from numerical analysis we can obtain its scaling dimension $\Delta_U$. The Callan–Symanzik function (23) in Figure 5 shows that the $\tau_j\tau_{j+1}$ perturbation at $C_1$ scales with $1/\nu = 0$, ie, is independent of the system size at the transition. From Equation (24) we conclude that the corresponding field has scaling dimension $\Delta_U = 2$ and is thus marginal.

The qualitative behaviour of the transition line close to $C_1$ is consistent with a simple mean-field argument. Decoupling the $U(\tau_j\tau_{j+1} + \text{H.c.})$ perturbation is tantamount to a shift in the on-site field term, $f \to f^* = f - 2\langle\tau_j\rangle U$; implying that the transition is shifted to $f_c = 1 + 2\langle\tau_j\rangle U$. Numerically we obtain $\langle\tau_j\rangle = 0.609 > 0$ at $U = 0$, in qualitative agreement with the positive slope of the transition between the trivial and topological phase.

---

[8]The scaling behaviour at the transition will be different in the $\mathbb{Z}_n$-symmetric model.

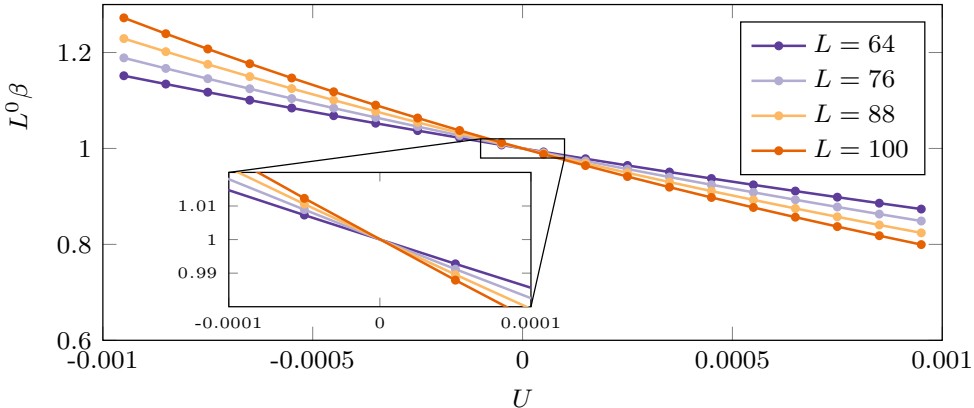

Figure 5: The Callan–Symanzik $\beta$ function for the $U(\tau_j \tau_{j+1} + \text{H.c.})$ perturbation at $C_1$. The scaling at the transition is independent of the system size, $\propto L^0$, indicating that the perturbation is marginal.

## 6.2 Potts transition in the vicinity of $C_2$

Finally, let us look more closely at the phase transition in the vicinity of $U = -1$. As already discussed in relation to (19), using a duality transformation the model with $f = 0$ can be written as two copies of a quantum Potts chain, implying that the transition at $U = -1, f = 0$ possesses central charge $c = 8/5$. Now let us reinstate the $f$-term within the dual description, which results in the Hamiltonian (again dropping the boundary terms; for details see Appendix A)

$$H = -\sum_{a=\text{o,e}} \left[ \sum_{j=1}^{L/2-2} \sigma_j^a (\sigma_{j+1}^a)^\dagger - U \sum_{j=1}^{L/2-1} \tau_j^a \right] - f \sum_{j=1}^{L/2-1} (\mu_j^\text{o} \mu_j^\text{e} + \mu_j^\text{e} \mu_{j+1}^\text{o}) + \text{H.c.} \qquad (26)$$

Starting from the $U = -1, f = 0$, the perturbing fields related to the lattice operators are known to be [60]

$$(U+1)(E^\text{o} + E^\text{e}), \quad 2f \mu^\text{o} \mu^\text{e}, \qquad (27)$$

which are the energy density and disorder fields respectively for each copies of the Potts chain. Both terms independently open up a gap, as can be seen in the phase diagram Figure 2. However, a proper combination of the perturbations will leave the system gapless, ie, there will be a gapless line $f_c(U)$. At first order in the couplings the renormalisation-group equations contain the scaling dimensions of the relevant fields $E^\text{o,e}$ and $\mu^\text{o,e}$

$$\partial_l (U+1) = (2 - \Delta_E)(U+1), \quad \partial_l f = (2 - \Delta_{\mu\mu})f = (2 - 2\Delta_\mu)f, \qquad (28)$$

with $\Delta_E = 4/5$ and $\Delta_\mu = 2/15$ [4]. At the phase transition neither flows to strong coupling, thus the scalings are necessarily proportional: $|f_c|^{\frac{1}{2-2\Delta_\mu}} \propto |U_c + 1|^{\frac{1}{2-\Delta_E}}$. Therefore, the transition follows a power law in the vicinity of $U_c = -1$ (see, eg, References [61,62] for a similar line of argument),

$$|f_c| \propto |U+1|^{13/9}. \qquad (29)$$

In Figure 6 we see that the numerically obtained transition points (black dots) are in very good agreement with the scaling prediction (red line). Thus the emerging picture is that under the perturbations (27) the $c = 8/5$ fixed point is unstable, with the flow along the line (29) being described by the Potts CFT with $c = 4/5$. This is also consistent with the fact that due to the $c$-theorem [4, 63] the central charge cannot increase under the renormalisation-group flow. We note in passing that such an analysis for the Ising transition in the ANNNI model shows similar behaviour, with the scaling exponent replaced by 7/4 [62].

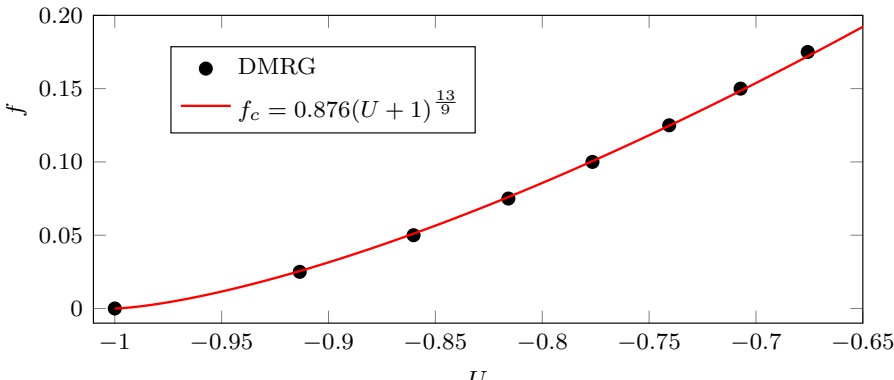

Figure 6: The phase boundary between the paramagnetic and topological phase. The dots are obtained with finite-size scaling from the DMRG calculation. The red line is the CFT prediction (29), with the prefactor obtained from a fit to the numerical data.

## 7 Lower half of the phase diagram ($f < 0$)

The phase diagram of ANNNI model is symmetric around the $f$-axis due to the underlying $\mathbb{Z}_2$-symmetry of the model. In contrast, the ANNNP model possesses a $\mathbb{Z}_3$-symmetry, which in turn breaks the symmetry of the phase diagram under $f \to -f$. While we have discussed above the phase digram in Figure 2 for $f > 0$, and seen that it looks very similar to the one of the ANNNI model, for $f < 0$ a completely different topography appears. It is the aim of this section to discuss the lower half of the phase diagram in detail.

### 7.1 Limit $f \to -\infty^+$: Effective XXZ model

We start the discussion by considering the limit $f \to -\infty$, in which the field term $-f(\tau_j + \tau_j^\dagger)$ in (14) becomes dominant. As the local eigenstates $|0\rangle_j, |1\rangle_j, |2\rangle_j$ have energies $-2f, f, f$, this limit projects onto the two local states $|1\rangle_j, |2\rangle_j$. This allows us to derive[9] an effective spin-1/2 model, with the third state, $|0\rangle_j$, only appearing in virtual processes.

The remaining terms in (14) are treated perturbatively. The first-order contributions to the effective Hamiltonian are (see Appendix C for the derivation)

$$H_{\text{eff}}^{(1)} = -\sum_j \left[ \sigma_j^+ \sigma_{j+1}^- + \sigma_j^- \sigma_{j+1}^+ + \frac{3U}{2}\sigma_j^z \sigma_{j+1}^z \right], \tag{30}$$

where $\sigma_j^\pm = (\sigma_j^x \pm i\sigma_j^y)/2$ with $\sigma_j^a$, $a = x, y, z$, denoting the Pauli matrices acting on lattice site $j$. Thus at leading order we recognise the spin-1/2 XXZ model with an U(1) symmetry generated by $\sum_j \sigma_j^z$. [We note that a similar argument was used in References [61, 64] to explain the appearance of critical $c = 1$ phases in parafermion chains to the XY phase of (30).] For this integrable model, the phase diagram is well-known [65] and consists of an antiferromagnetic Ising phase for $3U < -1$, a ferromagnetic Ising phase $3U > 1$, and a Luttinger-liquid phase with $c = 1$ in between. The Luttinger parameter of the critical phase is given by (at $f = -\infty$)

$$K = \frac{\pi}{2 \arccos(3U)}. \tag{31}$$

Note that the ferromagnetic Heisenberg point ($U = 1/3$) is not described by a CFT, as the dispersion becomes quadratic, or equivalently Luttinger parameter diverges. The transition

---

[9]We note in passing that the mapping to an effective two-state system seems possible for $\mathbb{Z}_n$-symmetric parafermion chains with $n$ being odd. However, we have not analysed the phase diagram in this case.

to the antiferromagnetic (AFM) phase at $U = -1/3$ appears at $K = 1/2$, where a gap opens due to the relevance of perturbations to the Luttinger-liquid field theory. However, the phase diagram obtained in this way, showing these three XXZ-phases, is not the complete picture. When we go slightly away from $f \to -\infty$, denoted as $f = -\infty^+$ in Figure 2, an additional $\mathbb{Z}_3$-phase emerges. This phase originates from the second-order contributions (see Appendix C for the details),

$$H_{\text{eff}}^{(2)} = \sum_j \left[ \frac{1}{6f}(\sigma_j^+ \sigma_{j+1}^- + \sigma_j^- \sigma_{j+1}^+) + \frac{1}{4f}\sigma_j^z \sigma_{j+1}^z \right. \tag{32}$$

$$\left. + \frac{1}{3f}(\sigma_j^+ \sigma_{j+2}^- + \sigma_j^- \sigma_{j+2}^+) + \frac{2}{3f}(\sigma_j^+ \sigma_{j+1}^+ \sigma_{j+2}^+ + \sigma_j^- \sigma_{j+1}^- \sigma_{j+2}^-) \right]. \tag{33}$$

In the absence of $U$, a similar expansion has been obtained in Reference [64] in the analysis of $S_3$-invariant spin chains [66]. In Reference [67], a similar effective description was found, discussing edge effects in fractional quantum Hall systems. The final term in (33) was also found in the effective study of Rydberg atoms in Reference [68].

The effect of the second-order contributions (32) is as follows: The first two terms only cause a redefinition of the XXZ parameters, which for example shifts the AFM transition to

$$U = -\frac{1}{3} + \frac{2}{9f}. \tag{34}$$

We note that the transition point is shifted to the left, in qualitative agreement with the numerical results leading to the phase diagram. In addition, the Luttinger parameter will also acquire corrections to the leading result (31). The two terms (33) need a more careful consideration: The first term is a next-nearest neighbour spin-flip term, conserving the U(1) symmetry. It has been shown that, for small perturbations, this terms only renormalises the XXZ parameters [69], leading to a further shift of the transition points on top of (34). The second term of (33) is more involved. It breaks the U(1) symmetry down to $\mathbb{Z}_3$. Within the bosonisation formalism this perturbation corresponds to a field with scaling dimension $\Delta_{+++} = 9/(4K)$ (see Appendix C.3). Whenever $\Delta_{+++} < 2$ this U(1)-breaking term is relevant, thus with (31) we see that at $f = -\infty^+$ a $\mathbb{Z}_3$ gapped phase should appear for

$$0.058 \approx \frac{1}{3}\cos\left(\frac{4\pi}{9}\right) < U < \frac{1}{3}. \tag{35}$$

These transition points will be shifted by $1/f$-corrections due to the corrections of the XXZ parameters originating from (32). We note in passing that a similar U(1)-breaking term has been shown to lead to $\mathbb{Z}_3$-order in a dilute Bose gas [70], although the precise relation to our setup remains unclear.

Finally we note that the analysis presented above critically depends on the absence of chirality breaking in the original model (14). Introducing a chirality-breaking term would, in the limit $f \to -\infty$, result in an additional, strong magnetic field term $\propto f \sum_j \sigma_j^z$ to be added to the effective Hamiltonian (30). This in turn would destroy the Luttinger-liquid phase as well as the antiferromagnetic and ferromagnetic Ising phases of the XXZ model and transform them into a trivial, paramagnetic phase.

In the following section we provide numerical evidence that the qualitative phase diagram deduced from bosonisation at $f = -\infty^+$ is also valid in the perturbative regime at $f = -30$. Furthermore, in Section 7.3 we show that the effective spin-1/2 description can be linked to the full ANNNP model even in the non-perturbative region at $f = -3$.

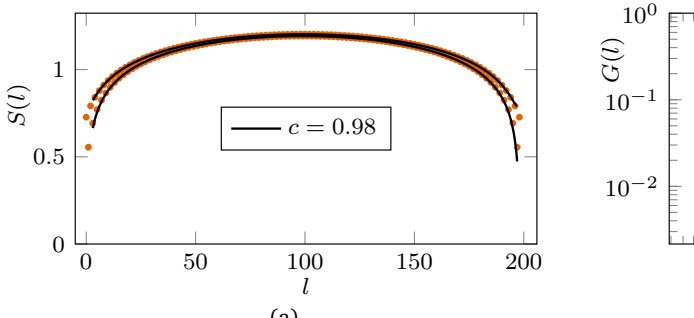
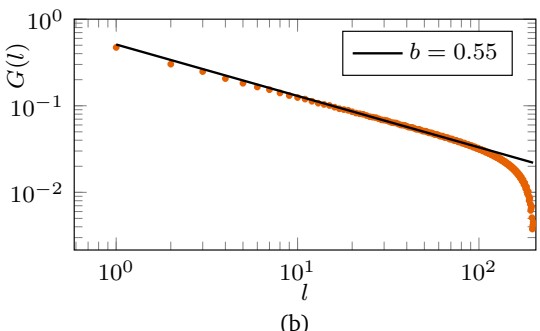

Figure 7: DMRG results (orange dots) for the $\mathbb{Z}_3$-ANNNP model for $U = -0.25$, $f = -3$ and $L = 200$. (a) Entanglement entropy together with the fitted prediction (36) (solid line). The alternation is the result of edge effects as was shown in Reference [71], see Equation (36). (b) Correlation function $G(l)$ with the corresponding power scaling $l^{-b}$ (solid line).

## 7.2 DMRG results: boundaries of the Luttinger-liquid phase

In principle, the DMRG simulations allow for a straightforward calculation of the central charge from the entanglement entropy via the Calabrese–Cardy formula (20). However, as the entanglement entropy of the XXZ model is sensitive to finite-size effects, a modified relation was proposed [71] taking the finite-size oscillations into account,

$$S_{\mathrm{mod}}(L,l) = S(L,l) + \frac{a \cos(\pi l)}{\frac{L}{\pi} \sin\left(\frac{\pi l}{L}\right)}. \tag{36}$$

Furthermore, we study the correlation function

$$G(l) = \left| \langle \sigma_j^\dagger \sigma_{j+l} \rangle \right| \propto l^{-b}, \quad b = \frac{1}{2K}, \tag{37}$$

for which we obtain the scaling exponent $b$ from the XXZ description. With the spin-1/2 projection we recognise

$$\langle \sigma_j^\dagger \sigma_{j+l} \rangle = \langle \tilde{\sigma}_j^+ \tilde{\sigma}_{j+l}^- \rangle \quad \text{with} \quad \tilde{\sigma}^\pm = \mathrm{diag}(0, \sigma^\pm). \tag{38}$$

The scaling behaviour then follows from standard bonsonisation [65].

As an example Figure 7 shows fits of these predictions to numerical results for $U = -0.25$ and $f = -3$, confirming $c \approx 1$ in the critical XXZ region as well as determining the Luttinger parameter to be $K = 0.9$.

To study the perturbative regime first, we take a cut along $f = -30$. For this cut, Figure 8(a) shows the central charge ($c$) and the scaling exponent ($b$). The differently coloured circles correspond to the full $\mathbb{Z}_3$-ANNNP model (red) and the various XXZ perturbative approximations (first order in dark blue, second order without the U(1)-breaking term in light blue, second-order with U(1)-breaking term in yellow). First of all, we note that the agreement of the results for the different models is remarkable except for the $\mathbb{Z}_3$-phase. This tells us that (i) the XXZ picture is a good approximation, (ii) the next nearest-neighbour spin-flip term, (33), denoted by "2$^{\text{nd}}$ no U(1)-br" is irrelevant as the results are indistinguishable from the "1$^{\text{st}}$" order plain XXZ results, (iii) for $0 \lesssim U \lesssim 1/3$ the U(1)-breaking term is important.

The central charge for the $\mathbb{Z}_3$-ANNNP model in the top panel of Figure 8(a) shows that there is a critical phase between gapped phases. However, the gapless region with $c > 0$



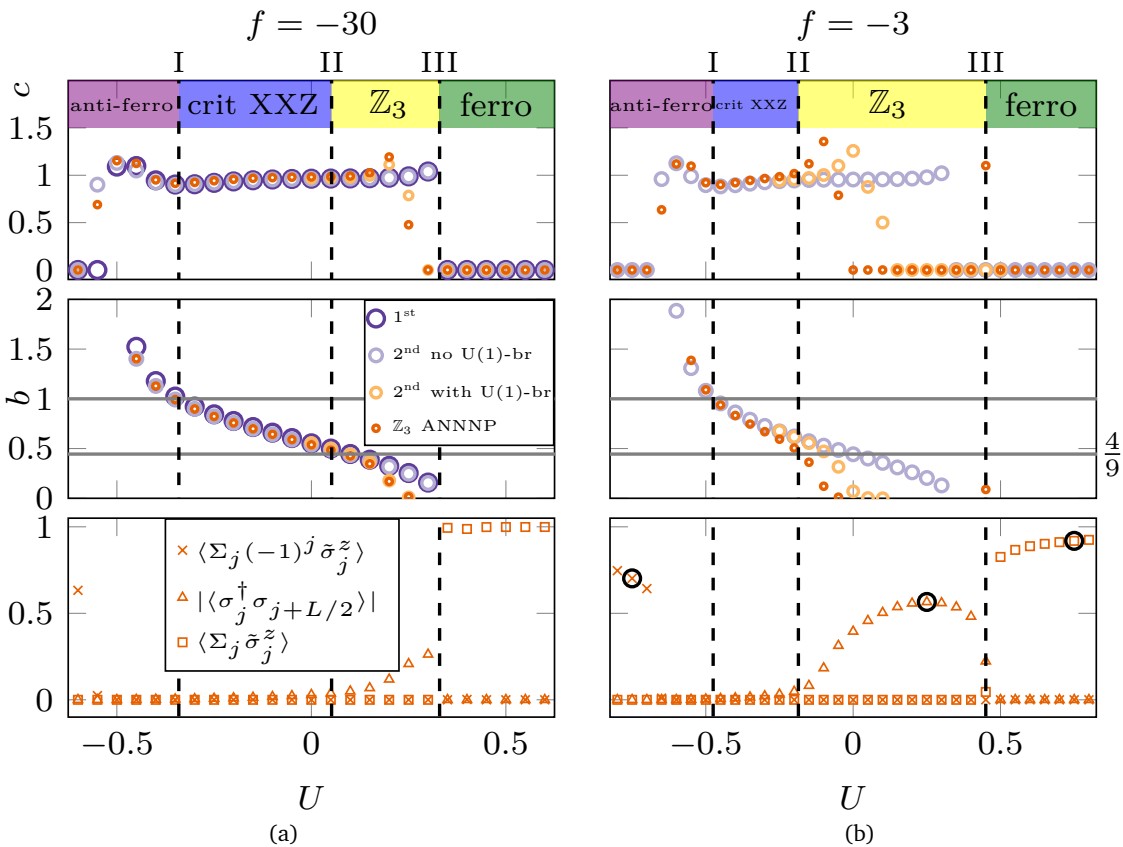

Figure 8: Numerical results for the different XXZ approximations (with and without the U(1)-breaking term) and the full $\mathbb{Z}_3$-ANNNP model for $L = 200$ for (a) $f = -30$ and (b) $f = -3$. The top and middle panels show the central charge $c$ and scaling exponent $b$, respectively; the solid horizontal lines at $b = 1$ and $b = 4/9$ indicate the values of the exponent belonging to transitions I and II. The bottom panels show the three order parameters introduced in (41), it will be discussed in Section 7.3. The black circles highlight the points for which further results are shown in Figure 11.

exceeds the regime denoted by "critical XXZ", which is estimated from values of the exponent $b$ shown in the middle panel. This deviation is the result of finite-size effects around the two transitions (I,II). Since both are described by a sine-Gordon term opening a gap in a Luttinger liquid, they are Kosterlitz–Thouless transitions (KTT) [65, 72]. The respective sine-Gordon terms responsible for transitions I and II are relevant for $K < 1/2$ and $K > 9/8$ respectively. Since the scaling of the correlation function (37) is related to $K$, we locate I at $b = 1$ and II at $b = 4/9$. As we see in the middle panel, the obtained transitions correspond accurately to the bosonisation predictions[10] (34) and (35) of $U_c^{\mathrm{I}} \approx -0.341$ and $U_c^{\mathrm{II}} \approx 0.0526$ respectively.

Let us have a closer look at transition II between the Luttinger-liquid phase and the $\mathbb{Z}_3$-phase. In contrast to the Potts transition discussed in Section 6, which was a second-order transition between to gapped phases, here we have to analyse a KTT between a gapped and gapless phase. For this it is known [65] that the gap closes extremely slowly and the observing the true transition point requires very large systems sizes. There are several studies [68, 73–79] addressing finite-size scaling for the KTT with the help of correlation length,

---

[10]Note that the value for $U_c^{\mathrm{II}}$ is slightly shifted from the prediction (35) because of the renormalisation of the Luttinger parameter $K$ due to (32).

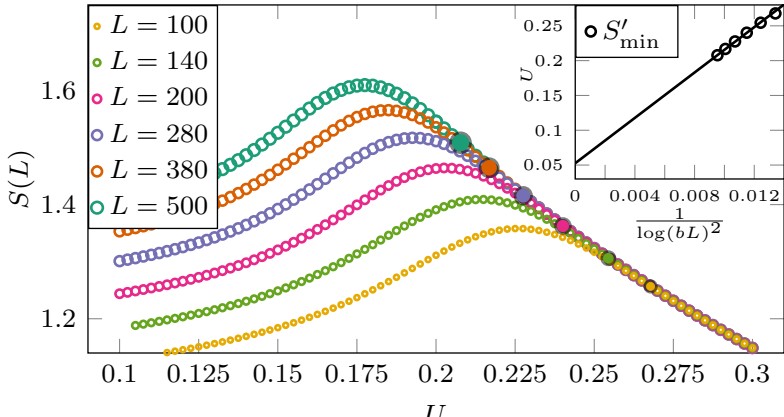

Figure 9: Central-cut entanglement entropy for $L = 100 - 500$ at $f = -30$, displaying the finite-size features close to the KTT between the $\mathbb{Z}_3$- and Luttinger-liquid phases. The inflection points are highlighted. The inset shows the finite-size scaling of the position of the inflection points, confirming the thermodynamic transition at $U_c^{\mathrm{II}} \approx 0.0526$ using (40).

fidelity and entanglement entropy. In our experience certain features of the central-cut entanglement entropy (CCEE) proved most useful in this case. Starting with (20) we define the CCEE as $S(L) \equiv S(L, L/2)$. Deep in the gapped $\mathbb{Z}_3$-phase the CCEE is independent of system size, $S(L) = \log(3)$. In contrast, in the critical phase it follows (20), $S(L) \approx \frac{\log(L)}{6}$. As the system approaches the critical region, there is a bump due to finite-size effects. This is shown in Figure 9, where CCEE is plotted for several system sizes for $f = -30$ as a function of $U$. We are not interested in the bump, but rather in the inflection point to the right of this bump (highlighted in Figure 9), in particular the position $U_{\mathrm{infl}}(L)$. Since the CCEE diverges with $L$ in the critical phase, we assume $U_{\mathrm{infl}}(L)$ to approach the true thermodynamic transition ($U_c$).

The finite-size scaling follows from the observation that

$$L \approx \xi \propto \exp\left(C/\sqrt{|U - U_c|}\right),\tag{39}$$

for a KTT, with $\xi$ being the correlation length which is cut off by the system size $L$, and $C$ some constant. We can rewrite (39) as

$$U(L) = U_c + \frac{a}{\log(bL)^2},\tag{40}$$

with $a$ and $b$ being some constants, and assume that features like the inflection point of the CCEE follow this scaling. The inset of Figure 9 shows that this assumption together with the predicted value $U_c^{\mathrm{II}} \approx 0.0526$ is indeed satisfied.

Applying the reasoning above at $f = -3$, ie, outside the perturbative region, we obtain from the scaling of the correlation function that $U_c^{\mathrm{I}} \approx 0.475$ and $U_c^{\mathrm{II}} \approx 0.185$ as shown in the middle panel of Figure 8(b). The latter is in agreement with the results from finite-size scaling of the CCEE, not shown in this paper. Moreover, we note that the effective spin-1/2 description deviates qualitatively around transition II. This is not surprising, since we have left the perturbative regime[11]. Nonetheless, even at $f = -3$ the local state $|0\rangle_j$ is (almost) projected out, hence the spin-1/2 interpretation is still reasonable.

---

[11]The numerics for the spin-1/2 model does show a (narrower) $\mathbb{Z}_3$-phase in between the Luttinger liquid and the ferromagnetic phase.

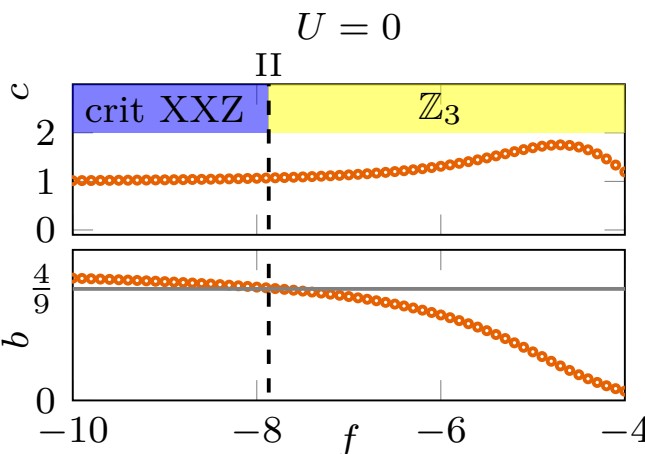

Figure 10: Numerical results for the $\mathbb{Z}_3$-ANNNP model for $L = 200$ at $U = 0$. The top and bottom panels show the central charge $c$ and scaling exponent $b$, respectively. The solid horizontal line at $b = 4/9$ indicates the value of the exponent belonging to transition II.

### 7.2.1 Chiral clock model at $U = 0$

In the absence of the $U$-term the model (14) becomes a special case of the chiral $\mathbb{Z}_3$-clock model [2], whose phase diagram as a function of the chiral angles $(\phi, \theta)$ was studied by Zhuang et al. [30]. More specifically, our model (14) at $U = 0, f < 0$ is equivalent to the chiral model at positive field strength and $\phi = \pi/3, \theta = 0$. The phase diagram for the latter shows a transition between the topological phase and a gapless, incommensurate phase with central charge $c = 1$. Using our conventions this translates into a transition from the topological phase to a gapless phase with $c = 1$ at $f \approx -4$. However, along the line $U = 0$ our results show a transition at $f_c^{\text{II}} \approx -7.87$. Both finite-size scaling of the CCEE and scaling of correlation function confirm this value. The latter can be seen in the bottom panel of Figure 10, where $b = 4/9$ signals the $\mathbb{Z}_3$-phase. We attribute the discrepancy with the estimated value $f \approx -4$ of Reference [30] to finite-size effects.

### 7.3 DMRG results: nature of the gapped phases

In order to further characterise the gapped phases, we have calculated three order parameters in the full ANNNP model. The results are shown in the bottom panels of Figure 8. Specifically, we determined the $\mathbb{Z}_3$-embedded antiferromagnetic and ferromagnetic order parameters as well as the long-range $\mathbb{Z}_3$-order defined as

$$\left\langle \sum_j (-1)^j \tilde{\sigma}_j^z \right\rangle, \quad \left\langle \sum_j \tilde{\sigma}_j^z \right\rangle, \quad G(L/2) = \left| \langle \sigma_j^\dagger \sigma_{j+L/2} \rangle \right|. \tag{41}$$

Here $\tilde{\sigma}_j^z = \text{diag}(0, 1, -1)_\tau$ in the local eigenbasis of $\tau_j$. In Figure 11 we show the order parameters for representative points in the different gapped phases, clearly confirming the nature of these phases as antiferromagnetic, ferromagnetic and $\mathbb{Z}_3$-long-range ordered, respectively. We note that the study of $G(L/2)$ is preferable over the short-range correlations $|\langle \sigma_j^\dagger \sigma_{j+1} \rangle|$, because the latter can be potentially close to 1 in the critical phase, while the long-range correlation decays with the system size (although with a power law). In contrast, $G(L/2)$ will become constant in the $\mathbb{Z}_3$-ordered regime, as can be seen exemplarily in Figure 11(b). We also note that the antiferromagnetic and ferromagnetic Ising phases have a two-fold degenerate ground state with a finite energy gap above; see Figure 14 in Appendix B.2. On the other hand, in the critical XXZ phase shows an even-odd effect (Figure 15, Appendix B.2).

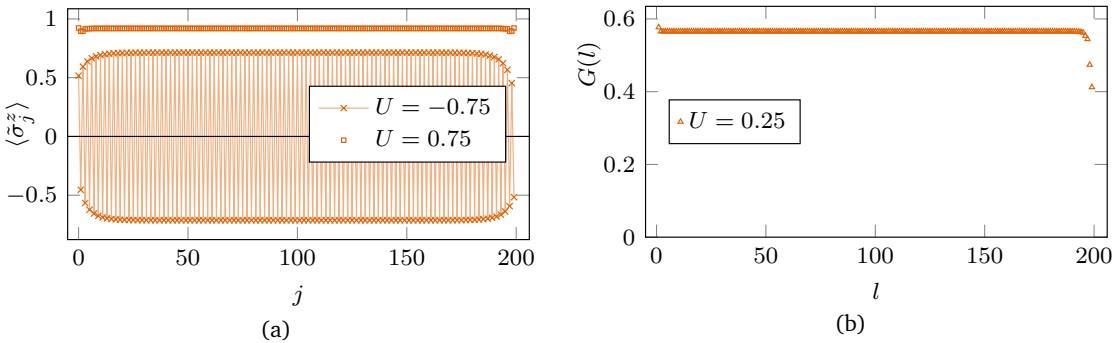

Figure 11: Example points displaying the respective order for the various gapped phases for $f = -3$. (a) The antiferromagnetic ($U = -0.75$) and ferromagnetic ($U = 0.75$) behaviour of the local magnetisation. (b) The correlation function (37) signalling $\mathbb{Z}_3$/topological order. All data for $L = 200$.

Coming back to the bottom panels of Figure 8 we see that the three gapped phases can be well distinguished by the order parameters (41). This reveals the nature of the phases and can be used to locate the phase transitions. In particular, the transition between the $\mathbb{Z}_3$-ordered and ferromagnetic phases becomes clear from the crossover in the respective order parameters.

Finally, from the topography of the phase diagram shown in Figure 12 in Appendix B.1 we deduce that the antiferromagnetic region extends up to vanishing $f$, while the transition to $\mathbb{Z}_3$-ordered phase keeps the ferromagnetic region from touching the $f = 0$ axis (in the studied range for $U$). We also infer, in combination with the discussion above, that XXZ critical region extends up to $U = -1, f = 0$ where the critical point with $c = 8/5$ is located. We would like to stress that the vicinity of this point is very difficult to study numerically, since both the left and right transitions are rather soft. Similar difficulties were experienced in the vicinity of a multi-critical point in the ANNNI model [62].

## 8 Intricate phases for $U > 1.5$

We have focused on the phase diagram of the model (14) for small to moderate values of $U$. In particular we identified a topological phase as well as gapped and gapless trivial ones. For the ANNNI model it is well known [10, 20–24] that at strong interaction strengths $U$ other phases (like a Mott insulating phase) exist. In analogy we expect the existence of intricate phases at strong values of $U$ in the ANNNP model as well.

A first idea can be obtained from the dual description (26) of the model. At $f = 0$ the model is equivalent (up to boundary terms) to two decoupled quantum Potts chains. Thus we expect a transition form the topological phase to a gapless phase with $c = 2$ at $U \approx 8$, which is consistent with preliminary numerical data. The coupling of the two Potts chains for $f \neq 0$ involves non-local terms in the dual description, thus intricate behaviour can be expected. The preliminary numerical data indicate the existence of several phases, including a critical XXZ phase showing even-odd effects in the system length. However, as the detailed analysis of this part of the phase diagram lies outside the scope of the present manuscript, we will leave it for future investigation [80].

# 9 Discussion

In this article we have studied an extended parafermion chain, which possessed terms coupling parafermions on four neighbouring sites. We mapped the model to the non-chiral $\mathbb{Z}_3$-ANNNP model via a Fradkin–Kadanoff transformation and analysed the phase diagram for weak to moderate couplings of the four-site term. By applying a combination of DMRG simulations, scaling arguments and analytical results in special limiting cases we identified four gapped phases: a topological phase possessing a three-fold degenerate ground state, a trivial (para-magnetic) phase as well as an antiferromagnetic and ferromagnetic Ising ordered phase. The latter two as well as an additional critical Luttinger-liquid phase can be connected to the well-known phase diagram of the XXZ Heisenberg chain. We provided evidence that the topological phase appears in between the Luttinger-liquid phase and the ferromagnetic Ising phase, and is due to the U(1)-breaking nature of the $\mathbb{Z}_3$-ANNNP model. Furthermore, we discussed a possible experimental realisation of the extended parafermion chain using hetero-nanostructures consisting of ferromagnets, superconductors and fractional quantum Hall states.

There are several directions for future studies: (i) Obviously the phase diagram for strong couplings $U$ of the four-site term could be analysed. For the interacting Majorana chain it is known [10, 22–24] that in the limit of strong interactions two additional phases exist, a Mott insulator and an incommensurate charge density wave. Our preliminary numerical data indicate that around $U \approx 8$ additional phases appear in the extended parafermion chain, so it would be interesting to analyse their properties and link them to the known results for the Majorana chain. (ii) The $\mathbb{Z}_3$-symmetry of the model (14) allows the inclusion of terms in addition to $U \sum_j (\tau_j \tau_{j+1} + \text{H.c.})$. For example, including a term $\sim \sum_j (\tau_j \tau_{j+1}^\dagger + \text{H.c.})$ allows the construction [42, 46] of a family of frustration-free models, of which the points $G_1$ and $G_2$ in Figure 2 are just special cases. These frustration-free models could serve as starting point for an analytic study of the topological phase. We note that the addition of such a term is also feasible within the framework of the hetero-nanostructures discussed in Section 3. (iii) The properties of the parafermion chain critically depend on the chirality breaking in the model, see, eg, References [30, 31, 44] for studies of the phase diagram in chiral parafermion chains. Thus it would be natural to extend the model (14) by including chirality breaking, which, as indicated in Section 7.1, is expected to have a drastic effect on the phase diagram.

# Acknowledgements

We thank Philippe Corboz and Paul Fendley for useful discussions. F.H. acknowledges support by the Deutsche Forschungsgemeinschaft (DFG, German Research Foundation) under Germany's Excellence Strategy – Cluster of Excellence Matter and Light for Quantum Computing (ML4Q) EXC 2004/1 390534769. H.K. was supported in part by JSPS Grant-in-Aid for Scientific Research on Innovative Areas No. JP20H04630, JSPS KAKENHI Grant No. JP18K03445, and the Inamori Foundation. This work is part of the D-ITP consortium, a program of the Netherlands Organisation for Scientific Research (NWO) that is funded by the Dutch Ministry of Education, Culture and Science (OCW).

# A Duality transformation

In this appendix we discuss the duality transformation of the Potts model (see, eg, Reference [81]), with extra care to treat the boundary terms. We start with the Hamiltonian (14)

$$H = -\sum_{j=1}^{L-1} \sigma_j \sigma_{j+1}^\dagger - f \sum_{j=1}^{L} \tau_j + U \sum_{j=1}^{L-1} \tau_j \tau_{j+1} + \text{H.c.}. \tag{42}$$

Let us now apply the following transformation,

$$\nu_j = \sigma_j \sigma_{j+1}^\dagger, \quad \mu_j = \prod_{i \le j} \tau_i^\dagger \qquad \Leftrightarrow \qquad \sigma_j = \prod_{i < j} \nu_i^\dagger, \quad \tau_j = \mu_{j-1} \mu_j^\dagger, \tag{43}$$

with auxiliary operator $\sigma_1 = \nu_0^\dagger$ and the exception $\tau_1 = \mu_1^\dagger$. Note that $\nu_L$ is not defined, which is not a problem for the moment. Applying this, the dual Hamiltonian reads

$$H = -\sum_{j=1}^{L-2} \nu_j - f \sum_{j=1}^{L-1} \mu_j \mu_{j+1}^\dagger + U \sum_{j=1}^{L-2} \mu_j \mu_{j+2}^\dagger + B + \text{H.c.}, \tag{44}$$

where $B = -\nu_{L-1} - f \mu_1^\dagger + U \mu_2^\dagger$. Up to boundary terms, for $U = 0$ we recognise that (42) and (44) are physically equivalent at $f = 1$, ie, the model is self-dual at this point in the thermodynamic limit. We note that the model (44) has been studied by[12] Zhang et al. [44] with a focus on the phase diagram in the presence of chirality breaking.

Next, we turn off the perpendicular field, ie, we consider $f = 0$. The operators $\nu$ and $\mu$ can be split on the odd/even (o/e) sites to obtain

$$H = -\sum_{a=\text{o,e}} \left[ \sum_{j=1}^{L/2-1} \nu_j^a - U \sum_{j=1}^{L/2-1} \mu_j^a (\mu_{j+1}^a)^\dagger \right] + B + \text{H.c.}, \tag{45}$$

where $B = -\nu_{L/2}^\text{o} + U(\mu_1^\text{e})^\dagger$ contains the boundary terms. We recognise two decoupled Potts chains in their dual representation: For each chain we can do another duality transformation

$$\tau_j^a = \mu_j^a (\mu_{j+1}^a)^\dagger, \quad \sigma_j^a = \prod_{i \le j} (\nu_i^a)^\dagger \qquad \Leftrightarrow \qquad \mu_j^a = \prod_{i < j} (\tau_i^a)^\dagger, \quad \nu_j^a = \sigma_{j-1}^a (\sigma_j^a)^\dagger, \tag{46}$$

with auxiliary operator $\mu_1 = \tau_0^\dagger$ and the exception $\nu_1^a = (\sigma_1^a)^\dagger$. This gives

$$H = -\sum_{a=\text{o,e}} \left[ \sum_{j=1}^{L/2-2} \sigma_j^a (\sigma_{j+1}^a)^\dagger - U \sum_{j=1}^{L/2-1} \tau_j^a \right] + B + \text{H.c.}, \tag{47}$$

with

$$B = -\left( \nu_{L/2}^\text{o} - \nu_1^\text{e} - \nu_1^\text{o} \right) + U(\mu_1^\text{e})^\dagger. \tag{48}$$

It is interesting to relate the original order parameter $\sigma_j$ to the new operators $\sigma_j^a$,

$$\sigma_j = \begin{cases} \prod_{i<(j-1)/2} (\nu_i^\text{o})^\dagger (\nu_i^\text{e})^\dagger & = \sigma_{(j-1)/2}^\text{o} \sigma_{(j-1)/2}^\text{e}, & j \text{ odd}, \\ \left[ \prod_{i<j/2} (\nu_i^\text{o})^\dagger (\nu_i^\text{e})^\dagger \right] (\nu_{j/2}^\text{o})^\dagger & = \sigma_{j/2-1}^\text{e} \sigma_{j/2}^\text{o}, & j \text{ even}, \end{cases} \tag{49}$$

with the inverse relation given by

$$\sigma_j^a = \begin{cases} \prod_{i<j} \nu_{2i-1}^\dagger = \prod_{i<2j} (\sigma_i)^{(-1)^i}, & a = \text{o}, \\ \prod_{i<j} \nu_{2i}^\dagger = \prod_{2<i<2j+1} (\sigma_i)^{-(-1)^i}, & a = \text{e}. \end{cases} \tag{50}$$

---

[12]The relation between the parameters in Equation (4) of Reference [44] and the ones in (44) is given by $h \to J \equiv 1$, $J \to f$ and $J' \to U$. In particular, the supercritical point corresponds in our convention to the limit $f = 2U$ with $U \to \infty$, indicating that the Potts transition between the trivial and topological phases extends to arbitrary large $U$.

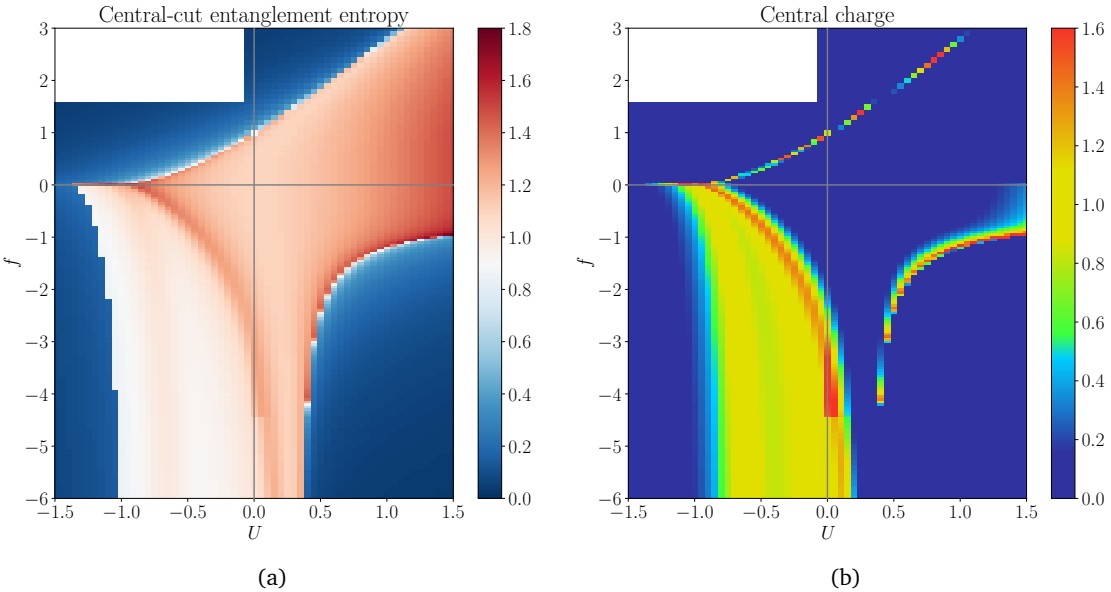

Figure 12: Rough topography of the phase diagram for the $\mathbb{Z}_3$-ANNNP model. The central-cut entanglement entropy and central charge results were obtained for small system sizes $L =$50–100 and low bond dimension in the parity sector 0. Even though the nature of the phases and transitions cannot be conclusively derived from these plots, it gives a good visual guide for the features to be studied in more detail.

Thus we see that the relation is non-local involving string operators.

We can also rewrite the symmetry operator, $\omega^P = \prod_j \tau_j = \mu_L^\dagger = (\mu_{L/2}^{\rm e})^\dagger = \prod_j \tau_j^{\rm e}$. For $f = 0$, the original Hamiltonian has another symmetry $\tilde{\omega}^P = \prod_j \sigma_j = \prod_{j,a}(\sigma_j^a)^\dagger$.

Finally, for completeness we can reinstate the $f$-term for the second transformation. Even though the resulting lattice model is non-local in terms of the original operators $\sigma$ and $\tau$, the following expression will be useful in Section 6.2

$$H = -\sum_{a=\rm o,e}\left[\sum_{j=1}^{L/2-1} \nu_j^a - U\sum_{j=1}^{L/2-1}\mu_j^a(\mu_{j+1}^a)^\dagger\right] - \sum_{j=1}^{L/2-1} f\mu_j^{\rm e}\left[(\mu_j^{\rm o})^\dagger + (\mu_{j+1}^{\rm o})^\dagger\right] + B + \text{H.c.}, \quad (51)$$

with $B = -\nu_{L/2}^{\rm o} - f\left[(\mu_1^{\rm o})^\dagger + \mu_{L/2}^{\rm o}(\mu_{L/2}^{\rm e})^\dagger\right] + U(\mu_1^{\rm e})^\dagger$.

# B Supporting numerical results

In this appendix we present additional numerical material to support certain points in the main text.

## B.1 Rough topography of the phase diagram

The overall structure of the phase diagram presented in Section 5 was determined largely based on an inexpensive DMRG calculation, ie, for small systems ($L = 50 - 100$). The results of these calculations are shown in Figure 12. It displays the central-cut entanglement entropy and central charge. The entanglement entropy follows naturally from the DMRG calculation.

With Schmidt decomposition we can write the ground state as

$$|\Psi\rangle = \sum_a s_a |\Psi_a^A\rangle |\Psi_a^B\rangle \,, \tag{52}$$

where $A, B$ are the left and right subsystems, such that $L_A + L_B = L$ and $\sum_a s_a^2 = 1$. Also $|\Psi_a^A\rangle$ and $|\Psi_a^B\rangle$ form an orthonormal basis in their respective subspace. The reduced density matrix becomes

$$\rho_A = \text{Tr}_B \rho = \sum_a s_a^2 |\Psi_a^A\rangle \langle \Psi_a^A| \,, \tag{53}$$

with the entanglement entropy given by [54]

$$S(L, L_A) = -\text{Tr}\rho_A \log(\rho_A) = -\sum_a s_a^2 \log(s_a^2)\,. \tag{54}$$

The area law predicts that the entanglement entropy should be constant with respect to system size for gapped systems, which can be used as a first tool to identify gapped phases studying the central-cut entanglement entropy $S(L, L/2)$.

As an example, consider the unique product state (15) the central-cut entanglement entropy is simply given by $S = -1 \log(1) = 0$. In the top left of the phase diagram in Figure 12(a) we find $S \approx 0$, indicating that this region is indeed connected to the trivial product state. On the other hand, for the $\mathbb{Z}_3$-ordered phase at $U = f = 0$, the ground state for each parity sector is a linear combination of the three degenerate ground states (17). Hence the central-cut entanglement entropy is given by $S = -\sum_{a=0}^{2} \frac{1}{3} \log(1/3) = \log(3) \approx 1.09$, which we observe throughout the topological phase. We note that the central-cut entanglement entropy can also be deceiving. For example, the ground states for the antiferromagnetic and ferromagnetic phases for $f < 0$ seem to be singly degenerate ($S = 0$), while they are in fact doubly degenerate with the two degenerate ground states lying in different symmetry sectors and the central-cut entanglement entropy vanishing in each of them.

In the critical regions the central-cut entanglement entropy is not a good indicator, since it diverges logarithmically with the system size. Instead, here we employ the central charge $c$ obtained by fitting the entanglement entropy (54) to the Calabrese–Cardy formula [56, 57] (20). It is important to note that this fit only give a qualitative view. The central charge in Figure 12(b) is often overestimated at points close to transitions, because at finite sizes the correlation lengths exceed the system size. Nevertheless, it shows the presence of a transition in the top left and bottom right. Moreover, there are several critical regions that can be identified, in particular the critical XXZ phase in the bottom left (see Section 7).

## B.2 Finite-size scaling of energy gaps

Here we present data for the finite-size scaling in the gapped regions discussed in Section 6 and 7. The two plots in Figure 13 show the gap for the paramagnetic/trivial phase and the $\mathbb{Z}_3$-ordered/topological phase. These confirm both the thermodynamic gaps as well as the respective degeneracies of the ground states.

In Figure 14 we show the finite-size scaling for the energy gap in the antiferromagnetic and ferromagnetic phase described in Section 7, showing that both are indeed gapped with a two-fold degeneracy.

On the other hand, in Figure 15 we see that both the gap to the first, second and third excited states vanish at $U = -0.25, f = -3$, thus this point indeed belongs to a critical region and not the three-fold degenerate topological phase. There is even-odd effect in the finite-size gap that we can explain from the effective XXZ description; for an extensive discussion see Reference [71]. For even chain lengths (and in the absence of a magnetic field) the ground

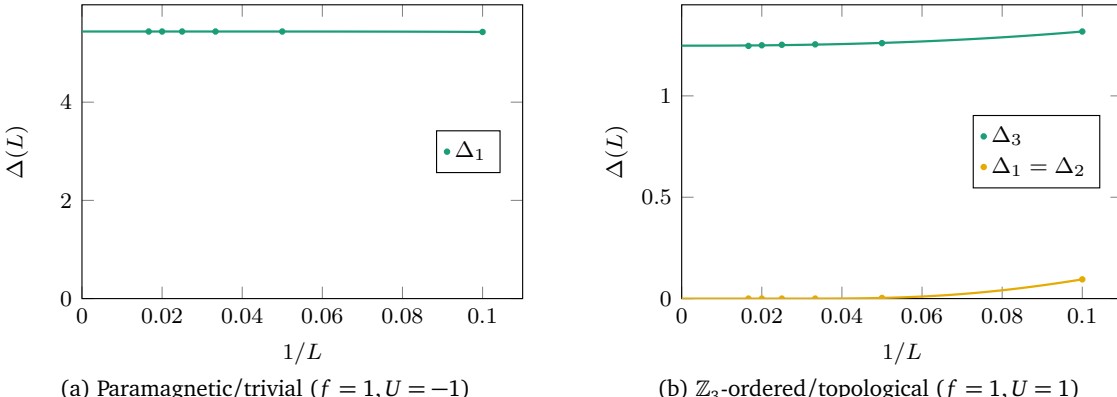

(a) Paramagnetic/trivial ($f = 1, U = -1$)  (b) $\mathbb{Z}_3$-ordered/topological ($f = 1, U = 1$)

Figure 13: Energy gaps $\Delta_n$ between the $n$th energy eigenstate and the ground state, obtained from finite-size scaling for system sizes $L = 10, 20, \ldots, 60$: (a) at $f = 1, U = -1$ in the paramagnetic/trivial phase, (b) at $f = U = 1$ in the $\mathbb{Z}_3$-ordered/topological phase.

state is unique with total spin $\langle S^z \rangle = \langle \sum_j \sigma_j^z \rangle = 0$. The first excited state is two-fold degenerate with $\langle S^z \rangle = \pm 1$, with the two states related by a global spin flip. On the other hand, for odd lengths the smallest magnetisation commensurate with the system is $\langle S^z \rangle = \pm \frac{1}{2}$, hence there is a double degeneracy of the ground states. We recognise this pattern in the finite-size scaling in Figure 15.

## C  Effective XXZ chains

In this appendix we derive the effective XXZ chain describing the limit $f \to -\infty$, which was presented in Section 7.1. We note that a similar expansion has been obtained in Reference [64].

### C.1  First-order term

The eigenvalues of the local field term $-f(\tau_j + \tau_j^\dagger)$ are $-2f, f, f$ for the eigenstates $|0\rangle_j, |1\rangle_j, |2\rangle_j$ respectively. Thus for $f \to -\infty$ there will be a large energy gap between the state $|0\rangle_j$ and the states $|1\rangle_j, |2\rangle_j$, which allows us to project onto a local, two-dimensional Hilbert space. Let us denote the resulting projected many-body Hilbert space by $\mathcal{G}$, with the notation $|\Psi_i\rangle \in \mathcal{G}$ and $|\Phi_i\rangle \notin \mathcal{G}$, and the respective energies due to this leading term by $E_{\Psi_i}$ and $E_{\Phi_i}$. For the remaining terms we can write down an effective first-order Hamiltonian describing the action of $V^\sigma = \sum_j v_j^\sigma$, with $v_j^\sigma = -\sigma_j \sigma_{j+1}^\dagger - \sigma_j^\dagger \sigma_{j+1}$, ie, terms that represent $\langle \Psi_i | V^\sigma | \Psi_k \rangle$. If we view the operators as tensor products of $2 \times 2$ matrices acting on the local states $|1\rangle_j, |2\rangle_j$, we can write

$$-\sigma_j \sigma_{j+1}^\dagger - \sigma_j^\dagger \sigma_{j+1} = -\sigma_j^+ \sigma_{j+1}^- - \sigma_j^- \sigma_{j+1}^+, \tag{55}$$

with $\sigma_j^\pm$ being the effective spin-1/2 raising and lowering operators acting at site $j$, ie, $\sigma_j^+ = |1\rangle_j \langle 2|_j$ and $\sigma_j^- = |2\rangle_j \langle 1|_j$. Similarly, we have $\sigma_j^z = |1\rangle_j \langle 1|_j - |2\rangle_j \langle 2|_j$. Using this we recognise that

$$\tau_j = \omega^{\sigma_j^z} = -\frac{1}{2} + i \frac{\sqrt{3}}{2} \sigma_j^z. \tag{56}$$

This allows us to rewrite the term $U \tau_j \tau_{j+1} + \text{H.c.}$ as

$$U \tau_j \tau_{j+1} + \text{H.c.} = -\frac{3U}{2} \sigma_j^z \sigma_{j+1}^z + \frac{U}{2}. \tag{57}$$

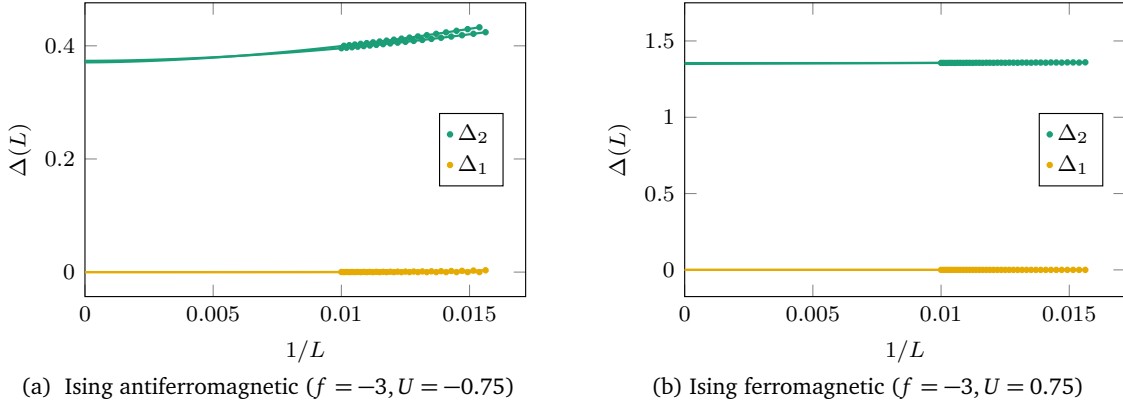

(a) Ising antiferromagnetic ($f = -3, U = -0.75$)      (b) Ising ferromagnetic ($f = -3, U = 0.75$)

Figure 14: Finite-size scaling of the energy gaps $\Delta_n$ for representative points in the Ising antiferromagnetic (a) as well as the Ising ferromagnetic (b) phase. Both DMRG results are for system sizes $L = 64, 65, \ldots, 100$. In both cases we find $\Delta_1 = 0$, showing that the ground states are two-fold degenerate, while $\Delta_2 > 0$ in the thermodynamic limit.

Taken together we thus deduce that at leading order the effective Hamiltonian describing the large $-f$ limit of the ANNNP model becomes

$$H_{\text{eff}}^{(1)} = -\sum_j \left[ \sigma_j^+ \sigma_{j+1}^- + \sigma_j^- \sigma_{j+1}^+ + \frac{3U}{2} \sigma_j^z \sigma_{j+1}^z \right]. \tag{58}$$

Hence the behaviour of the ANNNP model in this limit is governed by the XXZ Heisenberg chain, which is known to be critical for $|U| \leq 1/3$ with central charge $c = 1$ [65].

## C.2 Second-order term

The second-order terms originate from perturbations of the form

$$\sum_k \frac{\langle \Psi_i | V^\sigma | \Phi_k \rangle \langle \Phi_k | V^\sigma | \Psi_l \rangle}{E_\Psi - E_{\Phi_k}}, \tag{59}$$

where $E_\Psi = E_{\Psi_i} = E_{\Psi_l}$ the unperturbed energies of the ground states. Let us start with the contributions to effective two-site terms. The diagonal terms read

$$\frac{\langle 12 | v_j^\sigma | 00 \rangle \langle 00 | v_j^\sigma | 12 \rangle}{E_\Psi - E_{\Phi_k}} = \frac{\langle 21 | v_j^\sigma | 00 \rangle \langle 00 | v_j^\sigma | 21 \rangle}{E_\Psi - E_{\Phi_k}} = \frac{1}{6f}, \tag{60}$$

$$\frac{\langle 11 | v_j^\sigma | 20 \rangle \langle 20 | v_j^\sigma | 11 \rangle}{E_\Psi - E_{\Phi_k}} + \frac{\langle 11 | v_j^\sigma | 02 \rangle \langle 02 | v_j^\sigma | 11 \rangle}{E_\Psi - E_{\Phi_k}} \tag{61}$$

$$= \frac{\langle 22 | v_j^\sigma | 10 \rangle \langle 10 | v_j^\sigma | 22 \rangle}{E_\Psi - E_{\Phi_k}} + \frac{\langle 22 | v_j^\sigma | 01 \rangle \langle 01 | v_j^\sigma | 22 \rangle}{E_\Psi - E_{\Phi_k}} = \frac{2}{3f}, \tag{62}$$

which can be summarised as $\frac{1}{4f} \sigma_j^z \sigma_{j+1}^z + \frac{1}{2f}$. Similarly, the off-diagonal two-site contribution is given by

$$\frac{\langle 12 | v_j^\sigma | 00 \rangle \langle 00 | v_j^\sigma | 21 \rangle}{E_\Psi - E_{\Phi_k}} = \frac{1}{6f}, \tag{63}$$

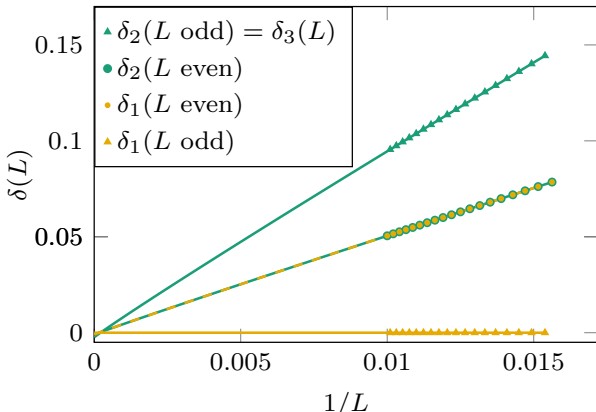

Figure 15: Finite-size scaling of the gap $\delta_n$ to the $n$-th excited states from system sizes $L = 64, 65, \ldots, 100$ for the system in the critical XXZ phase ($U = -0.25$, $f = -3$). The gap is depicted with a small $\delta$, to signal that it is zero in the thermodynamic limit.

which becomes the spin-flip term $\frac{1}{6f}(\sigma_j^+ \sigma_{j+1}^- + \sigma_j^- \sigma_{j+1}^+)$. Furthermore, there are three-site contributions such as

$$\frac{\langle 111|v_{j+1}^\sigma|102\rangle \langle 102|v_j^\sigma|222\rangle}{E_\Psi - E_{\Phi_k}} + \frac{\langle 111|v_j^\sigma|201\rangle \langle 201|v_{j+1}^\sigma|222\rangle}{E_\Psi - E_{\Phi_k}} = \frac{2}{3f}, \tag{64}$$

and its hermitian conjugate, which taken together become $\frac{2}{3f}(\sigma_j^+ \sigma_{j+1}^+ \sigma_{j+2}^+ + \sigma_j^- \sigma_{j+1}^- \sigma_{j+2}^-)$. This term breaks the U(1) symmetry of the XXZ chain, but preserves the $\mathbb{Z}_3$-symmetry. Finally, there is a next-nearest neighbour hopping term,

$$\frac{\langle 112|v_j^\sigma|202\rangle \langle 202|v_{j+1}^\sigma|211\rangle}{E_\Psi - E_{\Phi_k}} = \frac{\langle 122|v_{j+1}^\sigma|101\rangle \langle 101|v_j^\sigma|221\rangle}{E_\Psi - E_{\Phi_k}} = \frac{1}{3f}, \tag{65}$$

which can be written as $\frac{1}{3f}(\sigma_j^+ \sigma_{j+2}^- + \sigma_j^- \sigma_{j+2}^+)$. Taken together we arrive at the second-order Hamiltonian

$$H_{\text{eff}}^{(2)} = \sum_j \left[ \frac{1}{6f}(\sigma_j^+ \sigma_{j+1}^- + \sigma_j^- \sigma_{j+1}^+) + \frac{1}{4f}\sigma_j^z \sigma_{j+1}^z \right.$$
$$\left. + \frac{1}{3f}(\sigma_j^+ \sigma_{j+2}^- + \sigma_j^- \sigma_{j+2}^+) + \frac{2}{3f}(\sigma_j^+ \sigma_{j+1}^+ \sigma_{j+2}^+ + \sigma_j^- \sigma_{j+1}^- \sigma_{j+2}^-) \right]. \tag{66}$$

### C.3 U(1)-breaking term

In order to analyse the effect of the U(1)-breaking term within the bosonisation framework, we first bring the Hamiltonian (30) to its standard form. This is achieved by flipping the sign of the first two terms using the transformation $\sigma_j^\pm \to (-1)^j \sigma_j^\pm$. Now the bosonisation dictionary [65] shows that the low-energy behaviour of the XXZ model is governed by a Luttinger-liquid Hamiltonian

$$H = \frac{u}{2\pi} \int dx \left[ K (\nabla \theta(x))^2 + \frac{1}{K}(\nabla \phi(x))^2 \right], \tag{67}$$

with $\phi$ and $\theta$ being a bosonic field and its dual, and (at $f = -\infty$)

$$K = \frac{\pi}{2\arccos(3U)}, \quad u = \frac{1}{2 - 1/K} \sin\left[ \pi\left(1 - \frac{1}{2K}\right) \right] \tag{68}$$

denoting the Luttinger parameter and velocity respectively. The local spin-flip operators $\sigma_j^+$ are related to the bosonic field via [the Fermi momentum is given by $k_{\mathrm{F}} = \pi/(2a)$]

$$\sigma_j^+ = \sqrt{a}S^+(x) = \frac{e^{-\mathrm{i}\theta(x)}}{\sqrt{2\pi}}\Big[(-1)^x + \cos(2\phi(x))\Big], \tag{69}$$

where $x = ja$ with $a$ being the lattice constant (which we set to one), and $S^+(x)$ the continuum operator related to $\sigma_j^+$. Using (69) with the transformation $\sigma_j^\pm \to (-1)^j \sigma_j^\pm$ discussed above in mind, the U(1)-breaking term in $H_{\mathrm{eff}}^{(2)}$ becomes

$$(-1)^j \frac{2}{3f}(\sigma_j^+ \sigma_{j+1}^+ \sigma_{j+2}^+ + \sigma_j^- \sigma_{j+1}^- \sigma_{j+2}^-) \propto e^{-\mathrm{i}3\theta(x)} + \dots, \tag{70}$$

with the dots representing terms that are either less relevant or contain rapidly oscillating factors $(-1)^x$, which will thus not contribute in the continuum limit. Using the individual scaling dimensions $\Delta_{a,b} = \frac{a^2}{4K} + \frac{b^2 K}{4}$ of general vertex operators $e^{-\mathrm{i}a\theta(x)-\mathrm{i}b\phi(x)}$, we deduce that the scaling dimension of the U(1)-breaking term is given by

$$\Delta_{+++} = \frac{9}{4K}. \tag{71}$$

This shows that the U(1)-breaking term is relevant whenever $K > 9/8$.

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
