# Peer review of "Phase diagram of an extended parafermion chain"

_SciPost Physics Core, doi:SciPost Phys. Core 5, 008 (2022)_

## Round 1 · Referee Report · Anonymous (Referee 1) · 2021-8-23

Strengths

1- Timely and well written 2- The results are sound 3- The results are clearly and correctly summarized in the Abstract and in the Discussion 3- Rich in details, both in the main text and in the Appendices, which are very useful for the interested reader 4- The results are presented using a nice mixture of numerical simulations, analytical arguments and comparisons with previous results on similar systems

Weaknesses

1- The Introduction could be broader (and perhaps include more references). That would help the non-expert reader to better understand the scope and the importance of the work within the existing literature on parafermionic chains.

Report

The authors present a careful study of the phase diagram of a non-chiral Z3 parafermionic chain featuring weak-to-moderate coupling within four neighboring sites. The backbone of the analysis is represented by DMRG simulations, whose output is carefully analyzed and compared with analytical results available for some limiting cases. The range of validity of the results is clearly stated. Reasonable and interesting perspectives for future work are briefly discussed.

While this work is not groundbreaking and does not contain particular breakthroughs, it definitely advances the field and is based on solid and appropriate methods. I enjoyed reading the paper and I recommend its publication on SciPost Physics Core, as all the acceptance criteria are met.

Requested changes

I only have some minor comments/suggestions. I really hope that the authors will consider them in order to increase even more the high-quality of the manuscript.

1- As stated in the "Weaknesses" form, I find the one-page Introduction a bit short. That's especially true given the clarity of the other Sections, which makes them accessible also to a non-expert reader. The latter would definitely benefit from a couple of additional introductory and more general paragraphs. For example, it would be helpful to stress the importance of the topological phase in terms of the properties of the parafermion zero modes (e.g. their manipulation and its relation with quantum computation). Another point that could be better addressed is the generalization to Zn with n>3: so far it is only described as "straightforward" in a footnote, but I would try to explain this point a bit better with additional sentences and references.

2- Along the line of the previous point, I'd find it useful to add some more references and brief comments at the beginning in Section 3. For instance, I would have expected at least a reference to the recent preprint by Kim's and Yacoby's groups (https://arxiv.org/abs/2009.07836) on the experimental observation of induced superconductivity in the FQH bars. There are also several theoretical proposals that consider setups analogous to the one depicted in Fig. 1. Mentioning the related literature would help the reader in better understanding the relevance of such a setup and, therefore, also the importance of the present work.

3- Several numerical results concerning the values of central charge, critical exponents and other quantities, mainly discussed in Section 6.1.1, are "in good agreement" with the predicted ones. Just out of curiosity, what is the reason for the small mismatch? Is there a way to reduce it?

---

## Round 1 · Referee Report · Anonymous (Referee 2) · 2021-10-4

Strengths

  • The paper is well written, rich in details, and the methods very nicely outlined
  • The model and the parameter regime are physically motivated
  • The interplay between different methods reinforces the credibility of the results
  • The appendix is very helpful in guiding the reader and supporting the results
  • main results are sound and well outlined

Weaknesses

  • Some motivation regarding the study of parafermions (universal topological quantum computation) as well as some relation to experiments is missing

Report

The authors study the phase diagram of the $\mathbb{Z}_3$ anisotropic non-chiral (i.e. real coupling parameters) next-nearest neighbor Potts (ANNNP) model using analytical techniques and DMRG. The ANNNP is dual to an extended parafermion chain which could (maybe) be one day realized in experiments using fractional quantum hall- superconductors-ferromagnet heterostructures. The model is thus motivated by physical considerations, especially the origin of the the 4-site coupling term, which was not studied before, is motivated from cross-capacitance charging effects. The authors exhibit a good handle on the different techniques (duality transformations, links to CFTs etc), their range of validity, and give lots of details regarding the different phases. Effective theories (Luttinger liquid, c=4/5 CFT, Heisenberg XXZ etc) are given for relevant parameters. The interplay between the different methods reinforces the message and the credibility of the results.

The phase diagram of the model (in the relevant regime) is constructed in a clear, clever and detailed manner, with the appendix to guide the reader and support the results. I found the link between the ANNNP model and the XXZ Heisenberg model, with its corresponding higher order corrections, particularly interesting.

I learned a lot while reading the manuscript and the Appendix. The main result (phase diagram) of the paper is nicely depicted and the results well summarized.

As its fits the criteria of acceptance, I recommend publication on SciPost physics core.

I have a few comments/questions:

-The form of parameters $(J,f)$ as a function of the system (fig 1) parameters (ie SC and FM gap) can be in principle be derived, and they decrease exponentially as the separation between parafermions is increased. In general, tunneling across superconductors gives rise to complex coupling parameters $J$ (see Chen $\textit{et al}$, PRL. 116, 106405 (2016) as well as Groenendijk $\textit{et al}$ PRB 100, 205424 (2019)). The non-chiral limit (real parameters) corresponds physically to having set the chemical potential to zero throughout the system/suitable choice of gate voltage. Maybe the authors could comment on the link between the parameters of the physical system (superconducting gap, ferromagnet strength, gate voltage etc) and the $(f,J,U)$ parameters of the coupled parafermion chain, as well as what chirality breaking implies physically (vanishing chemical potential, choice of gate voltages etc). If one understands the U term as multiple tunneling of fractional charges between parafermions, I expect it to be exponentially suppressed in the distance between the four parafermions so that the limit $|U|<|f|$ makes sense. As this term corresponds to tunneling across both FM region and SC regions (see fig 1), out of curiosity, do you have an idea how it would look like as a function of the magnetic gap, superconducting gap, distance between PFs etc? Or how one could derive the form of the U parameter explicitly?

-One reason why people are interested in parafermions in themselves (rather than their Fradkin-Kadanoff duals, which are the Potts variable), is that they would be an important step towards universal topological quantum computations (See Dua $\textit{et al}$, PRB 100, 144508 (2019) on how to, in principle, obtain non-Clifford gates from parafermions, or also RevModPhys. 80, 1083 (2008) about the $\mathbb{Z}_3$ parafermionic Read and Rezayi state). I found no mention on quantum computation in the manuscript or as to why they are more interesting than Majoranas/$\mathbb{Z}_2$ Ising anyons, which the authors mention in the introduction. I think some mention about (universal) quantum computation would reinforce the relevance and the visibility of the manuscript.

-There have been a have been a few experiments towards realizing Pfs in FQH, (as in fig 1 of the manuscript): Wu $\textit{et al}$ (PRB 97, 245304 (2018)) managed to induce helical states in FQH, which is the main building block for parafermions. See also Wang $\textit{et al}$ (Nat Commun 12, 5312 (2021)) from the same group where the helical edge current in such a system is measured. Moreover superconductivity has been induced in the FQH regime in recent experiments in the groups of A. Yacoby and P. Kim (see arXiv:2009.07836). Mentioning these works would I think enhance the visibility of the manuscript to experimentalist: The ANNNP phase diagram obtained by the authors could maybe one day be used by experimentalists as a guide towards parafermions.

Regarding the phase diagram study I have a few questions:

  • Why are there two different fitted curves in the entanglement entropy fig. 7? Is this some kind of bifurcation or do they correspond to different values of $f$?
  • During duality transformations, boundary terms are often dropped, how (ir)relevant are they?
  • There seem to be some jumps in the order parameter and central charge in the anti-ferro case (fig 8 and 9), both for f=-30 and -3. Do you have any explanation for these jumps/discontinuities?

Requested changes

  • Referencing the aforementioned experimental papers may strengthen the relevance of the manuscript
  • The physical/experimental consequences of having a non-chiral model should in my opinion be mentioned (vanishing chemical potential), or how one could tune the $(J,f,U)$ parameters physically.
  • Some mention of why parafermions are interesting and their links with universal topological quantum computation would in my opinion reinforce the relevance and visibility of the manuscript.

---

## Round 2 · Referee Report · Anonymous (Referee 1) · 2022-1-12

Report

The authors implemented all the suggestions from the Referees. Importantly, in the revised manuscript, they corrected an error in Appendix C and its consequences. I find their analysis convincing and I agree that this revision does not affect the main results of the paper, whose quality and clarity remain intact.

Therefore, I still recommend its publication on SciPost Physics Core.

---

## Round 2 · Author Response

While working on the revisions we realised an error in our original line of argument related to the RG relevance of the U(1)-breaking term [Eq. (70) in the revised manuscript]. Due to the previously overlooked oscillating prefactor this term becomes relevant in the range (35). In addition we extended our numerical simulations [see, eg, new Figure 9] and their analysis and revised the manuscript to take these new results into account. We would like to stress that our main results (in particular the topology of the phase diagram and the existence of a topological phase) are not affected by these revisions, which only concern the phase diagram at very negative field strength f.

———————————
Reply to referee 1

We thank the referee for his/her very positive comments and helpful suggestions. To broaden the context, we have added some remarks in the introduction on the potential use of parafermions in quantum computation and experimental realisations. Furthermore, we added remarks on the generalisation of our results to general Z_n-symmetric models at suitable places. We also added a comment on the used bond dimensions in the DMRG simulations and its effect on the accuracy of the numerical data (see footnote 7).

———————————
Reply to referee 2

We thank the referee for his/her very positive comments and helpful suggestions. To broaden the context, we have added some remarks in the introduction on the potential use of parafermions in quantum computation. Furthermore, we extended the discussion of Section 3 regarding the parameters in our proposed experimental realisation, and added suitable references.

To answer the specific questions:
-The bifurcation is the result of finite-size effects as was shown in Reference [68]. We have added this remark in the caption of Figure 7.
-In connection with (19) we drop the boundary terms since we are only interested in the bulk behaviour. The derived predictions for the phase diagram along f=0 is consistent with our numerical results. For completeness we also discuss the boundary terms in Appendix A.
-We attribute the increase of the central charge above c=1 when leaving the AFM phase to the KT transition when going into the Luttinger phase, which generally makes the numerical simulations demanding.

---

## Round 2 · List of Changes

-added a sentence at the end of the first paragraph of the in introduction as well as the References 25,26
-added three sentences at the end of the second paragraph of the introduction as well as the References 35-40, both to add a discussion of quantum computation as suggested by referee 2
-revised a sentence after (2) to include Reference [43]
-added remarks on the generalisation to Z_n symmetry at several places
-added several point in Section 3 and adapted Figure 1 accordingly
-revised Figure 2 to incorporate the changes at very negative field strength f
-added the used bond dimension in the first paragraph of Section 5
-added footnote 6 on boundary terms appearing in the duality transformation
-added footnote 7 on the quality of the numerical data
-added a remark on the observed bifurcation of the data in the caption of Figure 7
-revised Section 7 regarding the relevance of the U(1)-breaking term and the resulting phase
-dropped the previous Section 7.4 on the almost frustration-free line, since the argument is replaced by the relevance of the U(1)-breaking term given in Appendix C.3
-slightly revised Section 8
-slightly revised the conclusion
-revised the acknowledgment
-slightly revised the wording in Appendix B and the caption of Figure 15
-revised the discussion of the U(1)-breaking term in Appendix C.3
-corrected a few typos

---

## Editorial Decision

published